# BRIDGING ML AND ALGORITHMS: COMPARISON OF HYPERBOLIC EMBEDDINGS

**Dorota Celińska-Kopczyńska & Eryk Kopczyński**
Institute of Informatics
University of Warsaw
`{dot,erykk}@mimuw.edu.pl`

## ABSTRACT

Hyperbolic embeddings are well-studied in the machine learning, network theory, and algorithm communities. However, as the research proceeds independently in those communities, comparisons and even awareness seem to be currently lacking. We compare the performance (computation time) and the quality of embeddings obtained by popular approaches as of 2025, both on real-life hierarchies and networks, and simulated networks. According to our results, the algorithm by Bläsius et al (ESA 2016) is about 100 times faster than the Poincaré embeddings (NIPS 2017) and Lorentz embeddings (ICML 2018) by Nickel and Kiela, while achieving results of similar (or, in some cases, even better) quality.

## 1 INTRODUCTION

An *embedding* is an instance of some mathematical structure contained within another instance, such as a group that is a subgroup. In general topology, embedding is a homeomorphism onto its image. Homeomorphisms are the isomorphisms in the category of topological spaces – they are the mappings that preserve all the topological properties of a given space. Given a network $(V, E)$, where $V$ is the set of vertices and $E$ is the set of edges, its embedding into some geometry $\mathbb{G}$ is a map $m : V \to \mathbb{G}$.

In hyperbolic geometry, all the postulates of Euclid hold, except for the *parallel axiom*. While parallel lines stay at a constant distance in Euclidean geometry, similar lines in hyperbolic geometry diverge exponentially. Recently, the area of *hyperbolic embedders* for networks –that is, algorithms for embedding networks into hyperbolic geometry– has gained popularity within the Machine Learning (ML) community. Those embedders exploit properties of hyperbolic geometry, such as exponential growth, making them a perfect match for visualizing and modeling hierarchical structures.

Probably the most influential paper (Nickel and Kiela, 2017) (*Poincaré embeddings*) shows that hyperbolic embeddings achieve impressive results compared to Euclidean and translational ones. The results have been improved even further in the follow-up (Nickel and Kiela, 2018) (*Lorentz embeddings*) by changing the used model of hyperbolic geometry. In the ML literature, those works are recognized as some of the first studies on hyperbolic embeddings (Gu *et al.*, 2019). However, it is worth noting that a rich history of hyperbolic embedding research precedes these papers. Hyperbolic embeddings were initially developed in the network theory (NT) community through the *Hyperbolic Random Graph* model (HRG) (Krioukov *et al.*, 2010). The algorithmic properties of this model, including embedding techniques, have been extensively studied in the algorithmic community. Surprisingly, there is limited cross-referencing between these research communities. For example, machine learning papers we have examined rarely cite algorithmic works, and vice versa. Also, we lack comparative studies that bridge those communities.

We believe the insights in the algorithmic/NT papers could significantly benefit the ML community. In this paper, we gather and experimentally compare 14 approaches from different communities using both real-world (38 networks, including 7 hierarchies, 21 connectomes, and 10 other networks) and simulated data (600 two-dimensional networks).

Against this background, our contributions are as follows:

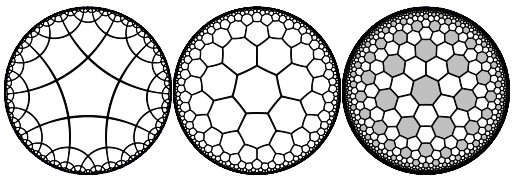

Figure 1: Tessellations of the hyperbolic plane. Bitruncated order-3 heptagonal tiling on the right.

- We present the first experimental comparison of hyperbolic embedders from the ML, NT, and algorithmic communities, establishing crucial connections among these research areas.

- We find that an $\tilde{O}(n)$ algorithm for creating hyperbolic embeddings (BFKL) (Bläsius *et al.*, 2016) that predates (Nickel and Kiela, 2017) is orders of magnitude faster while achieving results of comparable quality, or in some cases, better. Mercator embeddings (García-Pérez *et al.*, 2019) typically achieve intermediate-quality results while also being slow; TreeRep Sonthalia and Gilbert (2020) achieves good embedding quality on hierarchies but bad quality on networks. The recent embedder CLOVE (Balogh *et al.*, 2025) is also worth attention, both for its quality and time performance.

- While higher dimensions yield better embeddings according to standard quality measures (mAP, MeanRank, greedy routing success ratio, and efficiency), this is usually an artifact of optimization. Using information criteria principles, we introduce a new measure (Information Control Value, ICV). Unlike the standard measures, ICV penalizes embeddings with large radius and/or dimension, thereby enhancing the robustness of our comparisons.

## 2 THEORETICAL BACKGROUND

### 2.1 PRELIMINARIES ON HYPERBOLIC GEOMETRY

We start with the basics of hyperbolic geometry. For simplicity, we will focus on the hyperbolic plane $\mathbb{H}^2$, although the same ideas work in higher dimensions. See, e.g., the book (Cannon *et al.*, 1997) for a more thorough formal exposition, or the game HyperRogue (Kopczyński *et al.*, 2017) to gain intuitions. Recall the Euclidean space $\mathbb{E}^n$ is $\mathbb{R}^n$ with distance $\delta_E(x, y) = \sqrt{g_+(x - y, x - y)}$, where $g_+((x_1, \ldots, x_n), (y_1, \ldots, y_n)) = \sum_{i=1}^n x_i y_i$.

In modern terms, the simplest non-Euclidean geometry is spherical geometry. A two-dimensional sphere of radius 1 is $\mathbb{S}^2 = \{x \in \mathbb{R}^3 : g_+(x, x) = 1\}$. The distance is measured in terms of great circle arcs; a point in distance $r$ in direction (angle) $\phi$ from the central point $C_0 = (0, 0, 1)$ has coordinates $(\sin(\phi)\sin(r), \cos(\phi)\sin(r), \cos(r))$. The spherical distance between $x$ and $y$ can be computed as $\arccos(g_+(x, y))$; this is straightforward when $y = C_0$, and also true in general, since $g_+$ is invariant under the isometries (i.e., rotations) of the sphere.

Gaussian curvature is a measure of the difference of surface geometry from Euclidean geometry. A sphere of radius $R$, $R\mathbb{S}^2$, has constant Gaussian curvature $K = 1/R^2$. The hyperbolic plane is the opposite of spherical geometry, that is, it has constant negative Gaussian curvature. Hyperbolic surfaces are less ubiquitous, because they do not embed symmetrically into $\mathbb{E}^3$ – that would essentially require $R$ to be imaginary. However, they appear in nature when maximizing surface area is needed (e.g., lettuce leaves), and can be embedded symmetrically in the Minkowski spacetime. The hyperbolic plane $\mathbb{H}^2$ is thus $\{x \in \mathbb{R}^3 : x_3 > 0, g_-(x, x) = -1\}$, where $g_-$ is the Minkowski inner product $g_-((x_1, x_2, x_3), (y_1, y_2, y_3)) = x_1 y_1 + x_2 y_2 - x_3 y_3$ (the coordinate $x_3$ works like a time coordinate in special relativity). This is called the Minkowski hyperboloid model; many intuitions from spherical geometry work in this model, for example, a point in distance $r$ in direction (angle) $\phi$ from central point $C_0 = (0, 0, 1)$ has coordinates $p(r, \phi) = (\sin(\phi)\sinh(r), \cos(\phi)\sinh(r), \cosh(r))$. The hyperbolic distance between $x$ and $y$ can be computed as $\operatorname{arcosh}(g_-(x, y))$.

While the formulas of the Minkowski hyperboloid model tend to be intuitively obtainable by analogy to the sphere model, this model is not applicable to visualization, since it naturally lives in Minkowski spacetime rather than the usual three-dimensional space (we use Lorentz transformations rather than

Euclidean rotations for isometries involving the time coordinate). The most common method of visualization of the hyperbolic plane is the *Poincaré disk model*, first devised by Eugenio Beltrami, obtained as the stereographic projection of the Minkowski hyperboloid: $p(x, y, z) = (\frac{x}{z+1}, \frac{y}{z+1})$. This maps the (infinite) hyperbolic plane to a disk in the Euclidean plane. Figure 1 shows some tessellations of the hyperbolic plane in the Poincaré disk model. Each shape of the same shade in each of these tessellations is of the same size; the Poincaré disk model distorts distances so that the same hyperbolic distance appears smaller when closer to the boundary of the disk.

The Poincaré disk model is called a *model* (rather than *projection*) because it is often used directly, as an alternative representation of hyperbolic geometry. Many models are used; for us, the third important model is the *native polar* coordinates $(r, \phi)$. The formulas for converting from native polar coordinates to the hyperboloid model are given above as $p(r, \phi)$. All models describe the same (isometric) abstract metric space, so theoretically could be equivalently used in computations, although various models differ in how robust they are to numerical precision issues (as we will see later, hyperbolic geometry exhibits exponential growth, which makes such issues very significant (Celińska-Kopczyńska and Kopczyński, 2024b)). All can be generalized to higher dimensions and allow interpolation between possible values of curvature $K$. In our experience, people new to computational hyperbolic geometry use Poincaré model because introductory materials often focus on it; however, they have then difficulties computing distances and isometries, while such computations are straightforward in the hyperboloid model due to the full symmetry and spherical analogies. We see the difference between (Nickel and Kiela, 2017) and (Nickel and Kiela, 2018) as an example of this. The Minkowski hyperboloid is popular as the underlying model in the visualizations of hyperbolic geometry (Phillips and Gunn, 1992; Kopczyński *et al.*, 2017) due to simplicity and being a generalization of the *homogeneous coordinates* commonly used in computer graphics. The choice of the model may affect numerical precision (Floyd *et al.*, 2002; Celińska-Kopczyńska and Kopczyński, 2024b). As we will see later, native polar coordinates are commonly used for hyperbolic embeddings of social networks (Friedrich *et al.*, 2023).

## 2.2 HYPERBOLIC GEOMETRY IN VISUALIZATION, NT, AND ALGORITHMIC COMMUNITIES

While popular expositions of hyperbolic geometry usually focus on the sum of angles of a triangle being less than 180 degrees, what is actually important to us is exponential growth. As can be easily seen from the formula for $p(r, \phi)$, a hyperbolic circle of radius $r$ has circumference $2\pi \sinh(r)$; $\sinh(r)$ grows exponentially with $r$. This exponential growth, as well as the tree-like nature of the hyperbolic space, can be seen in Figure 1 and has found application in the visualization of hierarchical data, such as trees in the hyperbolic plane (Lamping *et al.*, 1995) and three-dimensional hyperbolic space (Munzner, 1998). Drawing a binary tree of large depth $h$ on Euclidean paper, while keeping all the edges to be of the same length, is difficult, because we eventually run out of space to fit all $2^h$ leaves. The hyperbolic plane, with its exponential growth, solves this issue perfectly.

This leads us to another application of hyperbolic geometry: the modelling of scale-free networks. Scale-free networks are commonly found in nature, technology, and social structures. They are characterized by the *power law* distribution of degrees (the probability that a random vertex has degree $\geq d$ is proportional to $d^{-\beta}$), as well as the high *clustering coefficient* (if node $a$ is connected to $b$ and $c$, the nodes $b$ and $c$ are also likely to be connected). Despite this ubiquity, it is not straightforward to find a mathematical model that exhibits both these properties. One such model is the *Hyperbolic Random Graph model* (HRG) (Krioukov *et al.*, 2010), characterized by parameters $N, R, \alpha, T$. In this model, $N$ nodes $\{1, \ldots, N\}$ are distributed randomly in a hyperbolic disk of radius $R$. Their angular coordinates $\phi$ are distributed uniformly, while their radial coordinates $r$ are distributed according to the density function $f(r) = \alpha \sinh(\alpha r)/(\cosh(\alpha R - 1)$. Let us denote with $m(i) \in \mathbb{H}^2$ the position of node $i$. Every pair of nodes $a$ and $b$ is then connected with probability

$$p(a, b) = (1 + \exp((\delta(m(a), m(b)) - R))/2T))^{-1}, \tag{1}$$

where $\delta(a, b)$ is the hyperbolic distance between the points in $\mathbb{H}^2$ representing the two nodes. The radial coordinates correspond to *popularity* (smaller $r$ = more popular) while the angular coordinates correspond to *similarity* (closer $\phi$ = more similar); the connections in a network are based on popularity and similarity. It can be shown that a random graph thus obtained has a high clustering coefficient and a degree distribution that follows a power law with $\beta = 2\alpha + 1$. There is extensive literature on the HRG model, including its algorithmic properties. Hyperbolic random graphs can be generated naively in $O(n^2)$ (Aldecoa *et al.*, 2015), in subquadratic time (von Looz *et al.*, 2015),

and in linear time (Bringmann *et al.*, 2019). Earlier works include (Kleinberg, 2007) and (Shavitt and Tankel, 2008). Despite being relatively popular, HRG is not the only generative model based on a similarity-popularity mechanism. Other approaches with similar properties include the earlier $\mathbb{S}^1$ model (Serrano *et al.*, 2008) allowing arbitrary degree distributions, GIRG (Bringmann *et al.*, 2018) in which the similarity space is a torus of some dimension $d$, generalized PSO (Papadopoulos *et al.*, 2012) in which additional *external edges* are added as the network grows, E-PSO (Papadopoulos *et al.*, 2015b), which uses only the external edges, and nPSO (Muscoloni and Cannistraci, 2018), which models realistic networks with communities using a non-uniform angular distribution.

With the theoretical generative models came the *embeddings* of real scale-free networks into the hyperbolic plane. An embedding of a network $(V, E)$ into geometry $\mathbb{G}$ is a mapping $m : V \to \mathbb{G}$. In (Boguñá *et al.*, 2010), such an embedding of the Internet was obtained and found to be highly appropriate for *greedy routing*. In greedy routing, a node $a$ wants to find a connection to another node $b$ by finding one of its neighbors $c$ which is the closest to $b$, then the neighbor of $c$ which is closest to $b$, and so on. Greedy routing is successful when we eventually reach $b$; in the *original* variant, it fails immediately when all the neighbors of $c$ are further away from $b$ than $c$; in the *modified* variant, such hops are allowed, and the method fails when we reach a cycle.

However, the embedding method used by Boguñá *et al.* (2010) required substantial manual intervention and did not scale to large networks (Krioukov *et al.*, 2010). Further research focused on finding unsupervised and efficient algorithms. An *embedder* is an algorithm that finds an embedding. While technically, any mapping is an embedding, we generally want the geometric structure of $m$ to be consistent with the structure of the network. *MLE embedders*, based on the maximum likelihood estimation (MLE) method from statistics, work by finding an embedding that maximizes the *loglikelihood* (LL). LL is the logarithm of the probability that if, for every pair of nodes $(a, b)$, we independently connect the nodes $a$ and $b$ with the probability computed according to the formula 1 (for some $R$ and $T$). Alternatively, *spring embedders* (Kobourov, 2013) simulate forces acting on the graph: attractive forces pulling connected nodes together, and repulsive forces pushing unconnected nodes away. Spring embedders have been adapted to non-Euclidean embeddings (Kobourov, 2013); however, the straightforward adaptation to hyperbolic geometry does not produce good embeddings of large radius (Bläsius *et al.*, 2016).

Note that embedding is a difficult computational problem – even computing LL according to the formula requires time $O(n^2)$, which is significant for large networks. The first algorithm for embedding large networks, HyperMap, worked in time $O(n^3)$ (Papadopoulos *et al.*, 2015b), later improved to $O(n^2)$ in HyperMapCN (Papadopoulos *et al.*, 2015a) and Wang *et al.* (2016a). Bläsius *et al.* (2016) developed a quasilinear algorithm for finding hyperbolic embeddings. This algorithm computes the HRG parameters based on the network's statistics. Then, it embeds the network in layers, starting from the nodes with the greatest degree, which form the center of the network. The algorithm, which we call the *BFKL* embedder, is evaluated on several scale-free networks from the SNAP database (Leskovec and Krevl, 2014) as well as randomly generated networks generated according to the HRG model. Eventually, Wang *et al.* (2016b) introduced a simple $O(n)$ algorithm based on hierarchical community detection Blondel *et al.* (2008), ordering the communities based on the *Community Intimacy* between pairs of communities, and basing the angular coordinates on this order and the radial coordinates on the degree. HMCS (Wang *et al.*, 2019) uses a similar approach, but changing a few details to obtain higher-quality embeddings.

Time complexity is just one facet of the quality assessment. We need some measures of the goodness-of-fit of an embedder. Embedders specialized to solve specific tasks popularized different measures. E.g., greedy routing performance is now commonly assessed using the stretch factor (GSF), greedy success rate (GSR), and greedy routing efficiency (GRE). The *stretch factor* (GSF) is the average ratio of the number of steps to the minimum possible (for successful paths). *Greedy success rate* (GSR) yields the share of the successful routings. Boguñá *et al.* (2010) showed that using greedy routing with the distances from the hyperbolic embedding achieves (GSR) 90%, which is significantly higher than, e.g., greedy routing based on actual geographical distances between the network nodes; Bläsius *et al.* (2016) found that greedy routing based on the BFKL embeddings again achieves good GSR. In (Bläsius *et al.*, 2018), the impact of numerical errors on the quality of hyperbolic embeddings and greedy routing is evaluated. *Greedy Routing Efficiency* (GRE) (Muscoloni *et al.*, 2017) is the average of $x/y$ over all pairs of nodes, where $y$ is the number of steps used by greedy routing and $x$ is the minimum possible; contrary to GSF, for failed routing we assume this ratio to be 0 (therefore, failed routings no longer can contribute positively to the measure). LL also became a

quality measure – good LL is achieved when connected nodes are placed close (distance less than $R$) and disconnected nodes are far away (distance greater than $R$). For tasks focused on angular positions (e.g., problems related to the similarity space, such as community detection), other measures, such as C-Score (Muscoloni *et al.*, 2017), became a standard. Other methods include mapping accuracy Zhang *et al.* (2021) and geometric congruence Cannistraci and Muscoloni (2022).

The official implementation of Bläsius *et al.* (2016) includes a spring embedder as a method of improving the result of the quasilinear algorithm; however, the running time of this step is $\Omega(n^2)$, which is too slow for large graphs. In (Celińska-Kopczyńska and Kopczyński, 2022), an alternative approach based on hyperbolic tilings, as shown in Figure 1 and previously used in HyperRogue (Kopczyński *et al.*, 2017), was introduced. The nodes of our graph are mapped not to points of the hyperbolic plane, but rather to the tiles of such a tiling. Also, the distances are computed in a discrete way, as the number of tiles. This is called DHRG, the *discrete* HRG model. This works, because such tilings' distances are a good approximation of hyperbolic distances (to a greater extent than similar approximations in Euclidean space (Celińska-Kopczyńska and Kopczyński, 2022)), and because the radii of HRG embeddings are large – the typical radii are on the order of $R = 30$ tiles of the bitruncated order-3 heptagonal tiling (1). One benefit of such a discrete representation is avoiding numerical precision issues. The other benefit is algorithmic: given a tile $t_1$ and a set of tiles $T$, we can compute an array $a$ such that $a[i]$ is the number of tiles in $T$ in distance $i$ from $t_1$ in time just $O(R^2)$. The time of preprocessing (adding or removing a tile from $T$) is $O(R^2)$ per tile. This gives us an efficient algorithm to compute the log-likelihood of a DHRG embedding, and also to improve a DHRG embedding in terms of LL by local search. Muscoloni *et al.* (2017) (Coalescent embedding) and García-Pérez *et al.* (2019) (Mercator) introduce ML algorithms to obtain or improve embeddings.

Most research concentrates on 2D embeddings, including the recent state-of-the-art, CLOVE (Balogh *et al.*, 2025), which arranges the communities using the algorithms for the Travelling Salesman Problem. Higher-dimensional embeddings have been studied recently (Bringmann *et al.*, 2019; Budel *et al.*, 2023; Kovács *et al.*, 2022; Jankowski *et al.*, 2023). A recent work (Celinska-Kopczynska and Kopczynski, 2024a) embeds into 3D Thurston geometries using tiles and simulated annealing.

## 2.3 HYPERBOLIC GEOMETRY IN ML COMMUNITY

Nickel and Kiela (2017) applied the Riemannian Stochastic Gradient Descent (RSGD) method to find hyperbolic embeddings. The algorithm is benchmarked on data that exhibit a clear latent hierarchical structure (the WordNet noun hierarchy) and on social networks (scientific collaboration communities). The quality is evaluated using new measures: MeanRank (MR) and Mean Average Precision (MAP). MR is the average, over all edges $u \to v$, of $r_{u,v}$, which is the number of vertices $w$ such that there is no edges from $u$ to $w$ and $w$ is closer to $u$ than $v$ (including $u$, not including $v$, thus MR $\geq 1$). MAP is the mean of average precision scores (AP) for all vertices. The average precision score of vertex $u$ is defined as $\sum_{i=1}^{k} i/r_{u,v_i}$, where $k$ is the number of vertices $v$ such that $u \to v$, and $v_i$ is the $i$-th closest of these vertices. In the case of WordNet, $u \to v$ iff $v$ is a hypernym of $u$; this is a transitive relation. In (Nickel and Kiela, 2018), the results are improved by using the hyperboloid model (referred to as the Lorentz model) instead of Poincaré model and evaluated using MR, MAP, and Spearman rank order on multiple real-world taxonomies, including the WordNet noun and verb hierarchies, the Enron email corpus, and the historical linguistics data.

While studies in NT on hyperbolic geometry seem to be inspired by the theoretical and applicational premises (using geometry as the means to understand nature), ML researchers quickly recognized the potential of including hyperbolic geometry as a part of an analytic pipeline, even in classification tasks (see, e.g., Chamberlain *et al.* (2017) application of hyperbolic embeddings to neural networks). That is why solutions to numerical precision issues have become a vibrant research area. Sala *et al.* (2018) studied the tradeoff between the number of dimensions and the number of bits used for representing the angles. They also gave a combinatorial method of embedding tree-like graphs. Yu and De Sa (2019) suggested a tiling-based model (LTiling) to combat the numerical precision issues. Their main idea is somewhat similar to DHRG, although while DHRG only uses the tiles, LTiling also includes the coordinates within the tiles. In TreeRep (Sonthalia and Gilbert, 2020), it is proposed that, instead of learning a hyperbolic embedding, we should instead learn a tree. Gu *et al.* (2019) embed networks not in $\mathbb{H}^n$, but in products of lower-dimensional spaces with hyperbolic, Euclidean, or spherical geometry, and in Guo *et al.* (2022), a method for visualizing higher-dimensional hyperbolic embeddings in $\mathbb{H}^2$ is proposed.

In Nickel and Kiela (2017), the early papers on hyperbolic visualizations ((Lamping *et al.*, 1995), but not (Munzner, 1998)) and the HRG model are cited, although the authors and reviewers seem not to be aware of the extensive literature on hyperbolic embeddings, including the paper (Muscoloni *et al.*, 2017) which uses ML methods and has appeared on arXiv in Feb 2016. The Poincaré embeddings are thus compared only to Euclidean and translational embeddings. This continues in the other papers mentioned in this section. As a result, many papers even directly claim or suggest that Nickel and Kiela (2017) were the first to consider hyperbolic embeddings, e.g., "Initial works on hyperbolic embeddings include Nickel & Kiela (2017) [...]" (Gu *et al.*, 2019).

We have found citations to NT research in Ganea *et al.* (2018); in Sonthalia and Gilbert (2020), Bläsius *et al.* (2016) is in the bibliography, but surprisingly, not referred to in text, despite the focus on speed; this paper also cites early work on hyperbolic embedding (Chepoi and Dragan, 2000), hyperbolic multi-dimensional scaling (Cvetkovski and Crovella, 2011), and embedding of $\delta$-hyperbolic graphs into trees (Chepoi and Dragan, 2000; Chepoi *et al.*, 2008; Abraham *et al.*, 2007). Comparisons between the results of different communities seem lacking.

## 3 OUR RESULTS

### 3.1 COMPARISON ON REAL-WORLD TAXONOMIES AND SCALE-FREE NETWORKS

For every network, we use the following experimental setup.

- Apply the following embedders to it: Poincaré embedding (PE) Nickel and Kiela (2017), Lorentz embedding (LE) Nickel and Kiela (2018), BFKL Bläsius *et al.* (2016), 2-dimensional and 3-dimensional *coalescent* embedder Muscoloni *et al.* (2017), HyperLink embedder (KVK) Kitsak *et al.* (2020), fast and full Mercator embedding García-Pérez *et al.* (2019), 3-dimensional Mercator embedding Jankowski *et al.* (2023), LTiling (Yu and De Sa, 2019), TreeRep (Sonthalia and Gilbert, 2020), Anneal (Celinska-Kopczynska and Kopczynski, 2024a), LPCS (Wang *et al.*, 2016b), HMCS Wang *et al.* (2019), CLOVE (Balogh *et al.*, 2025), DHRG embedding improvement Celińska-Kopczyńska and Kopczyński (2022) (applied to BFKL, PE, LE, and CLOVE).
- Evaluate the obtained embeddings according to quality measures from the literature: MAP, MR, GSF, GSR, GRE, and LL.

We also conduct analysis on hierarchies; in this case, we include the classic HypViewer tree embedder (Munzner, 1998) (if the hierarchy is not a strict tree, the parent is picked randomly) and do not evaluate on measures meaningful only for networks (GSF, GRE, and GSR). For all hierarchies, $u \rightarrow v$ iff $v$ is a superset (ancestor) of $u$; this is a transitive relation. We use the networks and hierarchies that have already been used as benchmarks in influential papers on hyperbolic embeddings. For the complete list of the networks and the hierarchies we used, see Appendix D.

An implementation of MR and MAP is available with Nickel and Kiela (2018). However, on larger graphs, some embedders (such as BFKL) generate embeddings of large radius. This implementation fails to evaluate such embeddings due to a numerical precision error. Therefore, we use our own implementation which avoids this issue. See Appendix B.

In the case of greedy routing measures, we prefer to use the original formulations, in which we immediately fail when there is no neighbor closer to the target. This is because some embedders use discrete tessellations, making it likely that some distances are equal. For original formulations, we can route randomly and use the expected route length (Celinska-Kopczynska and Kopczynski, 2024a). In the modified formulations, such an approach is ill-defined. Similar to (Celińska-Kopczyńska and Kopczyński, 2022), to aid comparisons, we report the LL values for the $R$ and $T$ values that maximize the log-likelihood (see Formula 1). We restrict our analysis to quality measures related to distance preservation; to our best knowledge, there are no measures that allow comparing the quality of angular positions in real-world embeddings.

The achievable quality of the embedding depends on the embedding dimension (achieving better results can be explained with higher dimensionality), therefore, in most cases, we compare 2D and 3D embeddings. (We include TreeRep because trees can be embedded into the hyperbolic plane.) For comparison, we also evaluate 5D PE, 50D, and 200D Euclidean embeddings (EE) (Nickel and

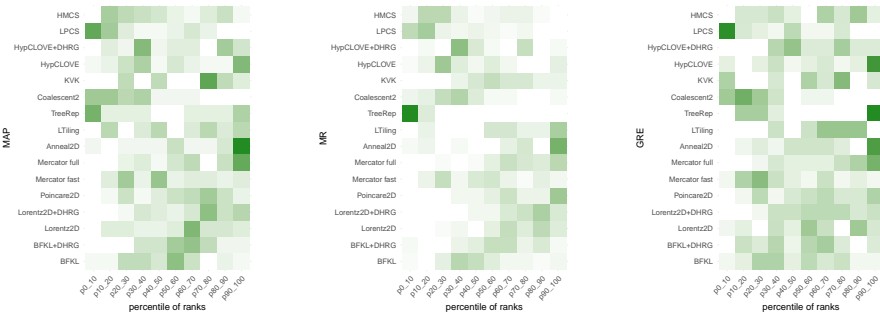

Figure 2: Quality assessment of embedders on real-world networks and hierarchies. Darker colors indicate that the given embedder occurred more frequently in the given percentile of ranks (higher percentiles are better) over all graphs benchmarked.

Kiela, 2017). Product space embeddings (Gu *et al.*, 2019) are an interesting approach, but they use higher-dimensional spaces, so they cannot be compared with 2D or 3D methods. The hMDS method from (Sala *et al.*, 2018) looks interesting, but it depends on the scaling factor, and it is not clear how to learn this parameter; therefore, we do not include this method in our experiments. Most embedders are randomized, so we have repeated a portion of experiments using different seeds; this does not usually change the rankings (Appendix H). We use the official implementations and hyperparameters; see Appendix A and the supplementary material.

Figure 2 shows the aggregate results, while details can be found in Appendices E (plots and tables) and I (NOUN hierarchy). Surprisingly, while BFKL has been designed specifically for scale-free networks and greedy routing, and LE has been benchmarked on hierarchies and MAP and MR, our results show that BFKL or DHRG achieves significantly better results on many hierarchies (BFKL: NOUN,VERBF,MESH; DHRG: mesh,tetrapoda), while Lorentz embeddings tend to achieve better results on networks, especially for greedy routing (better GSR and GSF). Still, the quality of BFKL, BFKL+DHRG, and 2D LE is similar across scale-free networks in our experiments, as measured by MR and MAP. One counterexample in the YEAST network, where BFKL achieves significantly better results than Lorentz on MAP (0.756 vs 0.542). In all cases, BFKL (and even BFKL+DHRG) is orders of magnitude faster, making LE impractical for larger graphs. The new CLOVE embedder tends to achieve even better results on hierarchies, in even better time. In many cases, DHRG is able to improve the results of fast embedders such as CLOVE while remaining reasonably quick.

HypViewer (Munzner, 1998) produces quite bad MR and MAP; however, it aims to put similar nodes close together, while due to how the transitive graphs are constructed for hypernymy hierarchies, high MR and MAP measures are achieved when similar categories (e.g., "lion" and "tiger") are closer to their hypernyms (feline, mammal, animal, entity) than to each other, which promotes longer edges on the outer levels of the hierarchy, and shorter edges in the center. The fast mode of Mercator usually produces worse embeddings than BFKL, while full Mercator usually achieves results between BFKL and 2D LE. Unfortunately, the full Mercator is slower than 2D LE for larger graphs. TreeRep is based on the idea of learning a tree instead of a hyperbolic embedding. We agree with this proposition for tree-like hierarchies, but for networks such as FACEBOOK and the connectomes, hyperbolic embeddings achieve significantly better results. (The hyperbolic plane is tree-like in large scale and Euclidean-like in small scale, and thus may potentially combine the advantages of both approaches). LTiling did not generally achieve better results than 2D LE in our experiments, while being significantly slower (contrary to DHRG, tiles are used only to improve numerical precision, not to make the process faster); however, this might be due to incorrectly set hyperparameters or testing on smaller, more shallow hierarchies, so the numerical precision issues did not yet become relevant. The results of LPCS are quite bad on connectomes, but on hierarchies and other networks, its results are comparable to BFKL. The coalescent embedder also performed relatively poorly in our experiments. The KVK embedder often achieved excellent results, but unfortunately turned out to be very slow – significantly slower than LE. Anneal works great for connectomes (which were its original area of application), but often turns out to be not that good for other data; this is probably

because Anneal can only produce embeddings of small radius, and connectomes, being physical networks, can have good embeddings of small radius.

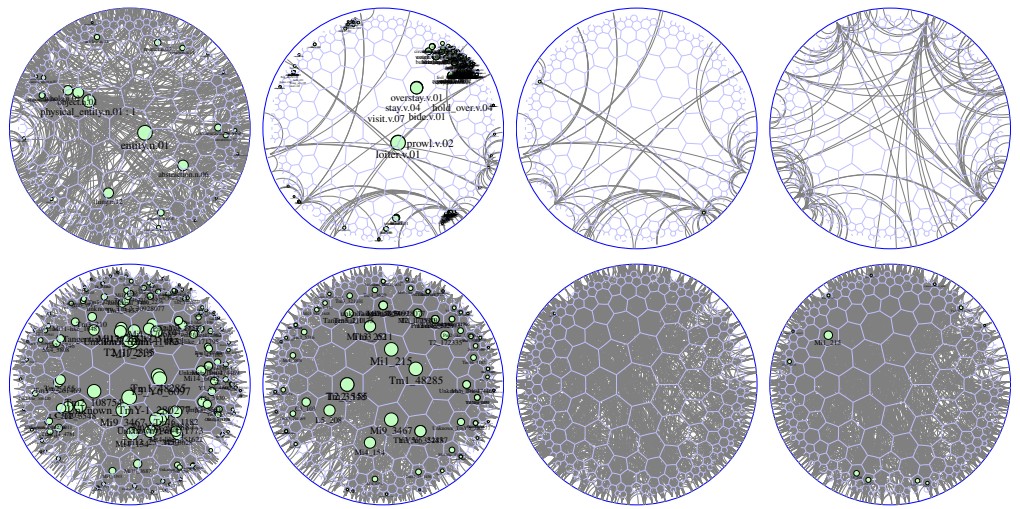

Figure 3: Top row: NOUN (Lorentz 2D). VERB (left to right: Lorentz 2D, Lorentz 2D+DHRG, BFKL). Bottom row: DROSOPHILIA1 (Lorentz 2D, Lorentz 2D+DHRG, BFKL, BFKL+DHRG).

## 3.2 VISUALIZATION

One application of 2D embeddings is visualization. We rendered the embeddings using the tools from DHRG; see Figure 3. All pictures are in Poincaré model, centered on the center of the hyperbolic disk used for embedding. One observation is that Lorentz embeddings tend to put nodes close to the center, while the center is generally avoided in BFKL, and DHRG improves the balance.

## 3.3 DIMENSIONALITY

According to all our experiments so far, higher-dimensional embeddings achieve better results than lower-dimensional ones. This result is trivially an artifact of optimization. Reducing the number of dimensions could be seen as imposing a restriction on that dimension; usually, optimization without restrictions yields better results. To make comparisons fairer, we need to use information criteria to control properly for this artifact. We introduce the *information control value* (ICV), based on the Minimum Description Length (MDL) principle (Rissanen, 1978), which takes into account both the

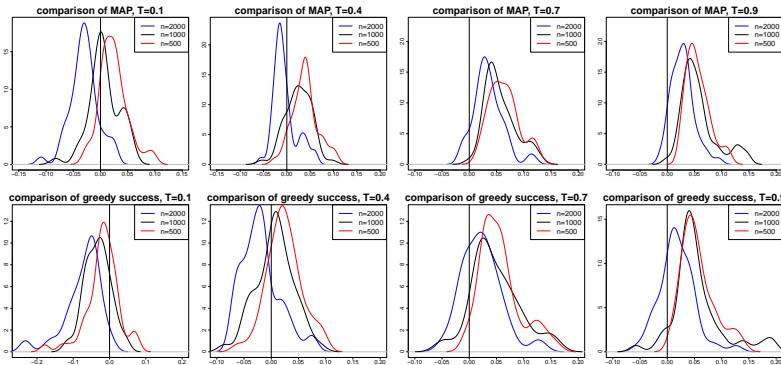

Figure 4: Density plots of the differences between the values of quality measures (MAP and GSR) obtained by Lorentz 2D and BFKL. Negative values indicate that BFKL performed better.

quality of edge prediction and the description length of the embedding; this description length is longer (worse) in more complex embeddings, such as those of higher dimension or radius. This is welcome, since more complex embeddings are harder to visualize, and also embeddings of higher radius are more prone to numerical errors (Bläsius *et al.*, 2018; Sala *et al.*, 2018; Celińska-Kopczyńska and Kopczyński, 2024b). According to our results, two-dimensional embeddings perform better for most real-world networks. The embedders we compare do not optimize the embedding radius, except Anneal, which enforces embeddings of small radii. To further improve ICV, we have also implemented a variant of DHRG that aims to reduce the embedding radius; the resulting improved BFKL is called **Penalty**. See Appendix C for the description of ICV and the Penalty approach.

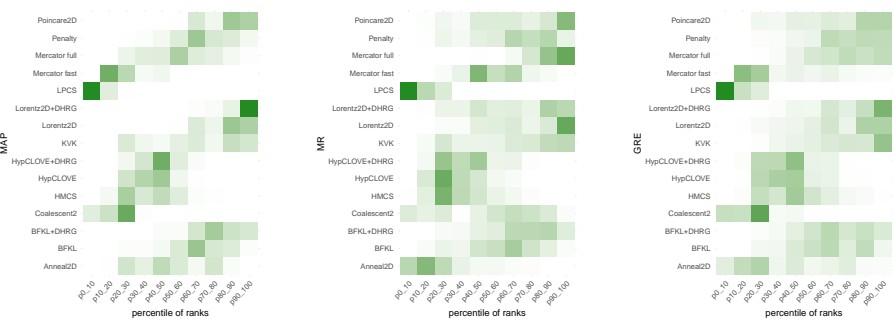

Figure 5: Quality assessment of embedders on simulated networks. Darker colors indicate that the given embedder occured more frequently in the given percentile of ranks (higher percentiles are better) over all graphs benchmarked.

### 3.4 COMPARISON ON ARTIFICIAL SCALE-FREE NETWORKS

For a more statistical analysis, we have also include comparison, especially BFKL and Lorentz 2D embeddings, on artificially generated scale-free networks. We use the generator from BFKL based on the HRG model, with default $\alpha = 0.75$, network sizes $n \in \{500, 1000, 2000\}$ and temperature $T \in \{0.1, 0.4, 0.7, 0.9\}$. This setup produces networks from the ultra-small world regime (Boguñá *et al.*, 2020). For such networks, the average distance grows slower than any polynomial of the logarithm of the network size, which is typical to heterogeneous networks with hubs. A significant share of real-world networks belongs to this regime, making it of greater interest to ML community.

Figure 5 depicts an aggregate ranking of all embedders. Regarding MAP and MR measures, we note apparent differences in the embedders' performance. LPCS tends to perform relatively poorly (usually in the bottom 10%), while 2D LE is significantly improved by discretization in the case of MAP. In contrast to the analysis of the real-world networks, CLOVE's performance is mediocre here – it occurs rarely within the top 10% of embedders. Regarding the greedy routing measures, we see little difference. Interestingly, LPCS and Coalescent embeddings tend to perform worse than other embedders on simulated networks generated from the HRG model. An analysis of possible explanations for this finding could constitute a future research line.

Fig 4 depicts the densities of the differences between the values of quality measures obtained by 2D LE and BFKL, and Table 1 contains results of the logit regressions on the determinants of the probability that BFKL would perform better than 2D LE in terms of a given quality measure. No matter the quality measure, according to our results, the greater the graph, the higher the probability that BFKL will perform better; however, with rising temperature, that probability decreases. Real-world networks are considered to have fairly large values of $T$, such as $T = 0.7$ used for Internet mapping (Bläsius *et al.*, 2016; Boguñá *et al.*, 2010), which is consistent with our results on real-world scale-free networks. Although our models were aimed at interpretation instead of prediction, we included information on the prediction quality, both from cross-validation and benchmarking. Our models are of satisfactory quality.

Even if our results suggest that, in many cases, 2D LE outperforms BFKL, it still comes at a high time cost. In Fig 6, we present the trade-off between the markup in time expenditure (how many times longer it takes to compute) in comparison to BFKL and the percentage gain in the quality of

|  | MAP | | GSR | | GRE | |
|---|---|---|---|---|---|---|
|  | Coeff. | Pr($>|z|$) | Coeff. | Pr($>|z|$) | Coeff. | Pr($>|z|$) |
| Intercept | -1.9701 | 7.09e-08 | 0.68510 | 0.007420 | 0.6402 | 0.01195 |
| Temp=0.4 | -0.8881 | 0.003800 | -2.1156 | 3.12e-12 | -2.1168 | 2.80e-12 |
| Temp=0.7 | -4.4173 | 5.20e-14 | -3.9175 | <2e-16 | -3.9746 | <2e-16 |
| Temp=0.9 | -4.4173 | 5.20e-14 | -4.3845 | <2e-16 | -4.3099 | <2e-16 |
| Size = 1000 | 1.6316 | 7.70e-05 | 0.92710 | 0.001930 | 0.8945 | 0.00287 |
| Size = 2000 | 3.7975 | <2e-16 | 2.53030 | 2.84e-15 | 2.6053 | 7.8e-16 |
| N | 600 | | 600 | | 600 | |
| $\text{ACC}_{cv}$ | 0.8835 | | 0.8231 | | 0.8235 | |
| $\text{ACC}_{bench}$ | 0.7867 | | 0.6200 | | 0.5356 | |
| $\kappa$ | 0.6165 | | 0.6133 | | 0.6233 | |

Table 1: Results of logit regressions for the determinants of BFKL embedder outperforming Lorentz 2D embedder in terms of quality measures. $\text{ACC}_{cv}$ and $\kappa$ are average accuracy and Kappa from 10-fold cross-validation; $\text{ACC}_{bench}$ is the accuracy of the naive model (always predict mode).

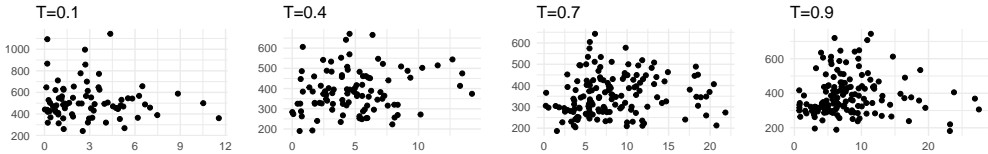

Figure 6: Comparisons of percentage gains in quality (MAP) of the 2D Lorentz embedding against the markup in time expenditure in comparison to BFKL embedder. X axis is the percentage gain in quality and the Y axis is how many times longer it takes.

the embedding (measured with MAP) resulting from using 2D LE. We conclude that there is no significant monotonic relationship between the time spent and the percentage gain in quality (p-values in Kendall-tau significance tests, as we encounter ties in our data that may make Spearman's rho inappropriate to use, are: 0.9867, 0.2143, 0.0106, and 0.0364 if we control for temperature 0.1, 0.4, 0.7, and 0.9 respectively. The last two results are insignificant at 1% significance level).

## 4 CONCLUSION

We compared the popular hyperbolic embedders from three communities, paying special attention to BFKL against 2D Lorentz embeddings (LE). Our main motivation is the apparent lack of awareness of the algorithmic results on hyperbolic embeddings in the ML community. Our results are of practical benefit: all the studied embedders generate static embeddings and thus, in ML pipelines, they can be easily replaced, to pick the best embedder based on the relevant quality measures and resource usage. In all experiments, BFKL runs significantly (about 100 times) faster than LE, while achieving results generally of similar quality. Higher-dimensional LE generally gets better results than both kinds of 2D embeddings, even in 3D; however, this no longer holds when we take information criteria into account. A more detailed study of our proposed criterion will be the subject of further research. In this study, we focus on ultra small world network regimes, generated with the HRG model. While we we do not perceive this limitation as a serious threat to validity of our results, further research could involve other regimes and other generative models such as PSO and nPSO. There are new hyperbolic embedders emerging each year; our framework can be easily extended to include them.

We have also found discrepancies between our results and the results in (Nickel and Kiela, 2017; 2018). In particular, in (Nickel and Kiela, 2017) 200D SGD Euclidean embeddings are performing worse than even low-dimensional Poincaré embeddings, but in our experiments, they consistently achieve significantly higher results (this particular case of non-reproducibility has been previously observed and studied in (Bansal and Benton, 2021)); in (Nickel and Kiela, 2018) Lorentz embeddings achieve significantly better results than Poincaré, while in our experiments, their performance is similar, and Poincaré is sometimes better. See Appendix G for details.

ACKNOWLEDGMENTS

We are grateful to our colleagues, the anonymous reviewers, and to the participants of Network Geometry: Theory and Applications workshop at NetSci 2025 for their valuable feedback, comments, and suggestions on the various stages of the preparation of this paper. This work has been supported by the National Science Centre, Poland, grant UMO-2019//35/B/ST6/04456.

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

## A  IMPLEMENTATION USED

The C++ implementation of our quality measures (MAP, MR, greedy routing measures, ICV) is published on GitHub as a part of the RogueViz engine (`https://github.com/zenorogue/hyperrogue`), released under GPL v3 (subdirectory `rogueviz/embeddings`). The exact version used in this paper, and the whole comparison framework (including auxiliary scripts), is available as the supplementary material. Due to the filesize limits on the supplementary material, we have also uploaded a larger version to `https://figshare.com/articles/dataset/Supplemental_material_for_the_paper_Bridging_ML_and_algorithms_comparison_of_hyperbolic_embeddings_ICLR_2026_/32228532`.

We have downloaded the embedders from the following repositories and use the following settings:

- Poincaré and Lorentz: `https://github.com/facebookresearch/poincare-embeddings` (last commit on Sep 16, 2021), Attribution-NonCommercial 4.0 International

  We use the hyperparameters `-epochs 1500 -negs 50 -burnin 20 -dampening 0.75 -ndproc 4 -eval_each 100 -fresh -sparse -burnin_multiplier 0.01 -neg_multiplier 0.1 -lr_type constant -lr 1 -train_threads 1 -dampening 1.0 -batchsize 50 -gpu 0` from the example `train-nouns.sh` from the repository, except that we requested using the GPU (`-train_threads 1 -gpu 0`). We also add the hyperparameters specifying a method (`-manifold poincare -dim 2`). For Lorentz 2D, Poincaré 3D, Poincaré 5D, Lorentz 3D, we replace `-lr 1` with `-lr 0.5 -no-maxnorm` (this setting comes from the suggestion about Lorentz embeddings in `train-nouns.sh`). The hyperparameters for SGD Euclidean embeddings are not given in the current official repository; we use the same parameters as for Poincaré (learning rate 1).

- BFKL: `https://bitbucket.org/HaiZhung/hyperbolic-embedder/overview` (last commit on Sep 8, 2016), no license given

  This method estimates the hyperparameters in `estimateHyperbolicParameters` method. We do not modify the original settings. The temperature ($T$) parameter for embedding is set to a low value 0.1 which should work well for embeddings, the parameter $\alpha$ is estimated based on fitting the power law, and the radius ($R$) is computed using a formula.

- DHRG: `rogueviz/dhrg` subdirectory in the RogueViz engine

  This method is parameterized by the tessellation used; we use the bitruncated order-3 heptagonal tiling. It does not create embeddings from scratch, but rather improves them using local search; we allow up to 110 iterations of local search. Local search computes the log-likelihood using the logistic function. We test it on the BFKL and Lorentz 2D embeddings. For the *penalty* variant, we set the parameter to 2 (see Appendix C), and perform 100 iterations of local search.

- TreeRep: `https://github.com/rsonthal/TreeRep` (last commit on Jun 23, 2023), GPL v3

  This method has no settings or hyperparameters (other than the number of threads, which we set to 8 as suggested in the repository).

- LTiling: `https://github.com/ydtydr/HyperbolicTiling_Learning` (last commit Mar 19, 2020), Attribution-NonCommercial 4.0 International

  We use the hyperparameters from the `-epochs 1000 -negs 50 -burnin 20 -dampening 0.75 -ndproc 4 -eval_each 100 -sparse -burnin_multiplier 0.01 -neg_multiplier 0.1 -lr_type constant -train_threads 2 -dampening 1.0 -batchsize 50 -manifold LTiling_rsgd -dim 2 -com_n 1 -lr 0.3 -no-maxnorm` from the `train-nouns.sh` example.

- HypViewer: `https://graphics.stanford.edu/~munzner/h3/download.html` (last modified in 2003), license in the COPYRIGHT file

  This method has no hyperparameters. For non-strict hierarchies we pick the parent randomly.

- Mercator: `https://github.com/networkgeometry/mercator` (last commit Jun 21, 2022), GPL v3

  Mercator has a setting for *fast* or *full* embedding; the *fast* method skips the likelihood maximization step. We apply both variants. We do post-processing of the inferred values of the radial positions. The parameter $\beta$ can be provided, but we use the default behavior, in which $\beta$ is inferred to reproduce the average local clustering coefficient of the original edgelist.

- d-Mercator: `https://github.com/networkgeometry/d-mercator` (last commit Nov 23, 2023), GPL v3

  The hyperparameters are similar to Mercator, except only *full* embedding is available for greater dimensions.

- Simulated annealing: `rogueviz/sag` subdirectory in the RogueViz engine

  This method is parameterized by the tessellation used; we use the bitruncated order-3 heptagonal tiling for 2D embeddings, and the subdivided(2) {4,3,5} honeycomb for 3D embeddings (the `g711` and `g435b2` settings from the original paper). As in the original paper, we set the parameter controlling the number of tiles to $M = 20000$. The number of iterations of simulated annealing is $N_S = 10000|V|$. As in the original paper, we run the embedder twice; the first pass is to obtain good initial values of the $R$ and $T$ parameters.

- Coalescent: `https://github.com/biomedical-cybernetics/coalescent_embedding` (last commit Jul 8, 2019)

  We use the hyperparameters and settings from `RUN_EXAMPLE.m`. Specifically, 2D embeddings uses RA1-LE-EA. 3D embeddings use RA1-ISO. We run the code in Octave (the free alternative of MatLab), which has no access to `graphallshortestpaths` function; we solve this issue by computing the table of shortest paths with our own C++ implementation.

- KVK: `https://bitbucket.org/dk-lab/2020_code_hyperlink/src/master/` (last commit Jun 11, 2011)

  This embedder has two parameters: the power-law exponent $\gamma$ and the temperature $T$. For simulated networks we use the actual temperature (providing more information to the embedder), whereas for real-world networks we use 0.1, similar to BFKL. As explained in the paper, $\gamma = 2\alpha + 1$; we take $\alpha$ estimated by BFKL.

- LPCS: the source code is included with the paper at `https://www.sciencedirect.com/science/article/pii/S0378437116000182`

  The source code is in MatLab. According to readme, before we call the MatLab function `main_LPCS`, we need to use the Fast Modularity Optimization (FMO) algorithm proposed by Blondel et al. to detect the hierarchical community structure, using the `multilevel.community` function in R (the name of this function is currently deprecated in favor of `cluster_louvain`); the source code of this part has not been included, thus we had to write our own, and also adapt the `main_LPCS` function to the situation when `cluster_louvain` returns a different number of community levels than 3. The code also has a hyperparameter `gama` (the power law exponent); we use the R function `fit_power_law(degree(g))` to estimate this exponent. We also had to fix some bugs (the code did not work when only two communities were found) and Octave warnings.

  Since the official implementation ran very slow in Octave, we have also reimplemented the algorithm in C++ (script/lpcs-remake.cpp). Our reimplementation fixes a bug in ConnectNextCom.m (which compares the intimacies of x(1) and x(2), while, according to the paper, intimacies of the first and last subcommunity in x should be compared here). In the following tables, embedding time is given for the original implementation, but not the reimplementation (the reimplementation is generally very fast).

- HMCS: We were unable to find an official implementation of this embedder, so we have edited our C++ reimplementation of LPCS to include the changes (obtaining the hierarchy by calling FMO algorithm repeatedly rather than just once; Community Closeness instead of Community Intimacy; angular size of a community based on the sum of degrees rather than the number of vertices). This embedder has one hyperparameter: the number of nested hierarchies to use. The authors suggest 2, or more for larger networks; we use 5 levels.

- CLOVE: `https://github.com/samu32ELTE/hypCLOVE` (last commit Oct 16, 2025)

We use the default values of all settings and hyperparameters: $\gamma$ to fit the degree distribution, degree fitting sample size of 100, automatically detected dendrogram, Leiden community detection method, exponential coarsening, the number 1 of anchor communities, Christofides algorithm for solving the Travelling Salesman Problem, `degree_greedy` node arrangement, community sector sizing based on the number of nodes in the community, and PSO radial coordinates assigned. For some hierarchies, the official implementation of CLOVE detects $\gamma < 2$ and fails with a parameter inference error. In these cases, we use $\gamma = 2$ instead.

For replicability we also control the PRNG seed.

We have downloaded the connectome datasets from `https://github.com/networkgeometry/navigable_brain_maps_data`. The scale-free networks are from the SNAP database (Leskovec and Krevl, 2014). The tree-of-life and GitHub followers graph dataset have been included with DHRG.

We use the following hardware:

[1] Intel® Core™ i7-9700K CPU @ 3.60GHz, NVIDIA GeForce GTX 1060 6GB/PCIe/SSE2, 96 GB RAM (we used zram for the embedders which did not fit in RAM)

[2] 11th Gen Intel® Core™ i7-11850H @ 2.50GHz, OpenGL renderer string: NVIDIA RTX A3000 Laptop GPU/PCIe/SSE2

Software: Arch Linux, g++ 12.2.1 to 15.2.1 (DHRG, BFKL, KVK, Anneal, Mercator), Julia 1.9.3 (TreeRep), Python 3.6 (Poincaré, Lorentz, ltiling, Mercator), Octave 10.3 (Coalescent, LPCS), R 4.5.2 (LPCS, creation of graphs)

The times reported in the paper have been obtained on [1]. Some experiments have been run on [2].

## B    COMPUTING DISTANCES, DISCRETE MAP AND MR

The distance between two points $p(r_1, \phi_1)$ and $p(r_2, \phi_2)$ in the hyperbolic plane can be computed as follows: (let $\phi = \phi_1 - \phi_2$)

$$
\begin{aligned}
\delta(p(r_1, \phi_1), p(r_2, \phi_2)) &= \delta(p(r_1, 0), p(r_2, \phi)) \\
&= \text{arcosh } g_-((\sinh(r_1), 0, \cosh(r_1)), \\
&\quad (\sinh(r_2)\cos\phi, \sinh(r_2)\sin\phi, \cosh(r_2)) \\
&= \text{arcosh } (\sinh(r_1)\sinh(r_2)\cos\phi + \cosh(r_1)\cosh(r_2)) \\
&= \text{arcosh } (\cosh(r_1 - r_2) + (1 - \cos(\phi))\sinh(r_1)\sinh(r_2))
\end{aligned}
$$

The last formula has better numerical properties (Bläsius *et al.*, 2016; Celińska-Kopczyńska and Kopczyński, 2024b). The distance formula in the Poincaré disk model can be computed similarly, although converting from Poincaré to hyperboloid needs solving a quadratic equation.

Still, the computation is somewhat slow: for each of $O(n)$ nodes, $O(n)$ distances from the other nodes need to be computed and sorted. It is possible to apply the discretization method from DHRG to quickly compute discrete analogs of MAP and MR (which we call dMAP and dMR). As mentioned in Section 2.2, discretization allows us to compute, for every node $t$, an array $a$ such that $a[i]$ is the number of tiles in $T$ in distance $i$ from $t$, in time $O(R^2)$. If $t$ has $e_t$ edges, we can compute a similar array $b[i]$ restricted to connected tiles in time $O(e_t R)$. Note that the formulas for MR and MAP given in Nickel *et al.* (2016) are for the case of continuous distances, and need to be adjusted for discrete values obtained from the DHRG model. In the case of MR, a non-edge with distance tie contributes 0.5 to $r_{u,v}$, and in the case of MAP, if there are $b[d]$ edges and $a[d]$ total nodes in distance $d$, we assume $k$-th of these edges to be ranked after $a[d](k - 0.5)/b[d]$ nodes. We can compute such MR and MAP knowing $a[i]$ and $b[i]$ for every node in total time $O(nR^2 + mR)$, where $m$ is the number of edges.

## C    CONTROL VALUE

This section describes the *information control value* (ICV), the embedding quality measure we propose. This value is based on the Minimum Description Length (MDL) principle (Rissanen, 1978). According to this principle, the shortest description of the data is the best model. We need $-\log_2(p)$ bit of information to describe an event happening with probability $p$.

In case of geometric embeddings, the description length consists of two parts: the description of the embedding itself, and the log-likelihood of obtaining the connections, given the embeddings. The second part is related to the log-likelihood used in the BFKL embedder. Recall that every pair of nodes $a$ and $b$ is then connected with probability $p(a, b) = p(\delta(a, b)) = (1 + \exp((\delta(a, b) - R))/2T))^{-1}$, where $\delta(a, b)$ is the hyperbolic distance between the points in $\mathbb{H}^2$ representing the two nodes. To compute the log-likelihood, we sum $\log(p(a, b))$ for every connected pair of nodes, and $\log(1 - p(a, b))$ for every unconnected pair of nodes. To compute the description length in bits, we use the same formula, except that we use $-\log_2(p)$ instead of the natural logarithm $\log(p)$. The parameters $R$ and $T$ are chosen in order to maximize the log-likelihood (equivalently, minimize the description length).

In a $d$-dimensional embedding, every node $i$ is described with coordinates $(r_i, \phi_i)$, where $r_i$ is the distance from the center, and $\phi_i \in \mathbb{S}^{d-1}$ is the angular coordinate. We assume that $r_i$ has exponential distribution $\text{Exp}(\lambda)$ restricted to $[0, R_{max}]$. We choose $R_{max}$ to be the maximum $r_i$ and $\lambda$ which maximizes the likelihood. Let $f_R$ be the density of this distribution of the radial coordinates $r_i$. For $\phi_i$, we assume that it is uniformly distributed in $\mathbb{S}^{d-1}$.

We assume that our coordinates are given with limited accuracy $\epsilon$. That is, instead of the precise $(r_i, \phi_i)$ obtained in our embedding, we use $(r'_i, \phi'_i)$ such that $p(r'_i, \phi'_i)$ is close to $p(r_i, \phi_i)$. To describe $r'_i$ such that $|r_i - r'_i| < \epsilon$, we need $-\log_2 \int_{r_i-\epsilon}^{r_i+\epsilon} f(r)dr$ bits. In case of angular coordinates, we need to divide the sphere of radius $r_i$, whose volume is proportional to $\sinh^{d-1}(r_i)$, by the area of $(d - 1)$ region of diameter $\epsilon$, which is proportional to $\epsilon^{d-1}$. Therefore, to describe $\phi'_i$, we need $d_i \cdot (\log_2 \epsilon - \log_2 \sinh(r_i))$ bits (as long as $r_i > \epsilon$). Since we know the positions of nodes $a$ and $b$ with error $\epsilon$, in the formula for $p(a, b)$ we take not $p(\delta(a, b))$, but $p'(a, b) = \frac{1}{2}(p(\delta(a, b) - \epsilon) + p(\delta(a, b) + \epsilon))$.

For the given $\epsilon$, we obtain the total description length $L$ as a sum of description lengths of $r_i'$, $\phi_i'$ for all nodes $i$, and $p'(a,b)$ for all pairs of nodes $(a,b)$. We choose the $\epsilon$ which minimalizes this description length $L$. Generally, a smaller $\epsilon$ increases the description length for $r_i'$ and $\phi_i'$, but decreases the description length of $p'(a,b)$.

To normalize the total description length, we compare $L$ with the description length $N$ of the naïve non-geometric representation, which simply assigns the same connection probability $p$ to every pair of nodes. Theoretically, a good geometric represesentation should obtain $L < N$; however, some of the algorithms we study obtain $L > N$. If there are $n$ nodes and $m$ edges, the minimum description length $-m \log_2 p - (\binom{n}{2} - m) \log_2 (1 - p)$ is obtained for $p = m/\binom{n}{2}$. Our *control value* is then $N/(N + L)$. This value is bounded from below by 0 (the worst case $L = \infty$) and from above by 1 (which would be obtained for $L = 0$). Good geometric representation achieve the control value of at least $\frac{1}{2}$, which corresponds to $L = N$.

Note that, for $d_1 < d_2$, a $d_1$-dimensional hyperbolic embedding $e_1$ can be considered a $d_2$-dimensional embedding, simply by considering the $\mathbb{H}^{d_1}$ that $e_1$ uses as a subspace of $-bbH^{d_2}$. All quality measures found in the literature and studied in this paper (log-likelihood, MAP, MR, GSR and GSF) will give exactly the same result whether $e_1$ is considered $d_1$-dimensional or $d_2$-dimensional; in other words, these measures will always give advantage to higher-dimensional embedding methods. In contrary, the *control value* will penalize higher-dimensional embeddings, as the part of the description length which corresponds to $(\phi_i)$ will be larger in higher dimension. Furthermore, *control value* will also penalize embeddings of larger radius. This is welcome, since embeddings of large radius are harder to visualize, and also more prone to numerical errors (Bläsius *et al.*, 2018; Sala *et al.*, 2018; Celińska-Kopczyńska and Kopczyński, 2024b).

In the Penalty variant of BFKL+DHRG, a node placed in distance of $\delta$ steps from the center of the model costs $K \cdot \log(r_\delta)$, where $r_\delta$ is the number of tiles in distance $\delta$. Instead of optimizing only the log-likelihood, we optimize the sum of log-likelihood and this cost. For $K = 1$ this cost corresponds to the part of description used to describe the angular coordinate. In our experiments we take $K = 2$ to make the embeddings even smaller.

# D    REAL-WORLD HIERARCHIES AND NETWORKS USED

Table 2 contains the list of real-world hierarchies and networks we benchmark all the embedders on. In the case of VERBF, we had to add an extra root node, since BFKL requires the network to be connected. We have not included other networks used in BFKL benchmarks because they are too large for slower algorithms such as Poincaré and Lorentz embeddings. In LE, the Enron email corpus and the historical linguistics data are analyzed using weighted edges, so we cannot compare them to BFKL or DHRG.

# E    RESULTS ON REAL-WORLD HIERARCHIES AND NETWORKS

The detailed results of our evaluation on real-world hierarchies can be found in Table 3. We also include MAMMAL (the mammal subtree of Noun). The detailed results of our evaluation on real-world networks can be found in Tables 4,5. Figures 7, 8, 9, 10, 11, and 12 contain visualizations of MAP, MR, GSR, GSF, and ICV on those hierarchies and networks. Figure 13 shows the aggregate information for the remaining measures.

Note that some algorithms are very slow, making them not feasible to run on large graphs. We do not provide the results in these cases.

Regarding ICV, we observe that Anneal2 yields very good results on most real-world networks except the Macaque4 connectome. This is because Anneal2 uses a fixed small embedding radius (about 7.9) which is usually smaller than for other embedders, yielding good ICV. Macaque4 is a connectome with only 29 nodes, which makes other embedders use small radius (2.13 for BFKL, 3.37 for Mercator, 6.73 for LPCS, HMCS and Clove). Furthermore, multiple embedders achieved perfect score on routing-based measures (i.e., 100% success rate and efficiency) on the Macaque4 connectome.

| name | type | details | $\|V\|$ | $\|E\|$ | source |
|---|---|---|---|---|---|
| ias | Internet | autonomous systems | 23748 | 58414 | IM (Boguñá *et al.*, 2010) |
| facebook | social | social circles | 4039 | 88234 | BDsG (Leskovec and Krevl, 2014) |
| followers-2009 | social | GitHub followers | 74946 | 537952 | D |
| openflights | transport | transport network | 3397 | 38460 | MC (OpenFlights website) |
| grqc | citation | general relativity | 4158 | 13422 | PSTY (Leskovec and Krevl, 2014) |
| astroph | citation | astrophysics | 17903 | 196972 | P (Leskovec and Krevl, 2014) |
| condmat | citation | condensed matter | 21363 | 91286 | P (Leskovec and Krevl, 2014) |
| hepph | citation | high-energy physics | 11204 | 117619 | P (Leskovec and Krevl, 2014) |
| yeast | biology | yeast metabolism | 1458 | 1948 | STY (Jeong *et al.*, 2001) |
| diseasome | biology | disease relationships | 516 | 1188 | ST (Goh *et al.*, 2007) |
| noun | hierarchy | WordNet | 82115 | 743086 | PLSTY (Miller, 1994) |
| acm | hierarchy | ACM classification | 2114 | 8121 | L |
| mammal | hierarchy | WordNet | 1180 | 6540 | pY (Miller, 1994) |
| verbf | hierarchy | WordNet | 13543 | 48621 | LY (Miller, 1994) |
| mesh | hierarchy | hierarchy | 58737 | 300287 | L (Rogers, 1963) |
| tetrapoda | hierarchy | hierarchy | 11262 | 527580 | (Maddison *et al.*, 2007) |
| csphd | hierarchy | hierarchy | 1025 | 3978 | STG (De Nooy *et al.*, 2018) |
| CElegans | cell | nervous system | 279 | 2287 | sMAT Varshney *et al.* (2011) |
| Drosophila1 | cell | optic medulla | 350 | 2887 | MA Shinomiya *et al.* (2022) |
| Drosophila2 | cell | optic medulla | 1770 | 8904 | A Shinomiya *et al.* (2022) |
| Mouse2 | cell | retina | 916 | 77584 | A Helmstaedter *et al.* (2013) |
| Mouse3 | cell | retina | 1076 | 90811 | A Helmstaedter *et al.* (2013) |
| ZebraFinch2 | cell | basal-ganglia (Area X) | 610 | 15342 | A Dorkenwald *et al.* (2017) |
| Macaque1 | area | cortex | 94 | 1515 | A Kaiser and Hilgetag (2006) |
| Macaque2 | area | cortex | 71 | 438 | A Young (1993) |
| Macaque3 | area | cortex | 242 | 3054 | A Harriger *et al.* (2012) |
| Macaque4 | area | cortex | 29 | 322 | A Markov *et al.* (2013) |
| Cat1 | area | cortex | 65 | 730 | A Scannell *et al.* (1995) |
| Cat2 | area | cortex and thalamus | 95 | 1170 | A Scannell *et al.* (1999) |
| Cat3 | area | cortex | 52 | 515 | A Scannell *et al.* (1999) |
| Human1 | area | cortex | 493 | 7773 | A Hagmann *et al.* (2008) |
| Human2 | area | cortex | 496 | 8037 | A Hagmann *et al.* (2008) |
| Human6 | area | whole brain | 116 | 1164 | A Gray Roncal *et al.* (2013) |
| Human7 | area | whole brain | 110 | 965 | A Gray Roncal *et al.* (2013) |
| Human8 | area | whole brain | 246 | 11060 | A Gray Roncal *et al.* (2013) |
| Rat1 | area | nervous system | 503 | 23029 | A Bota and Swanson (2007) |
| Rat2 | area | nervous system | 502 | 24655 | A Bota and Swanson (2007) |
| Rat3 | area | nervous system | 493 | 25978 | A Bota and Swanson (2007) |

Table 2: Our benchmark graphs. 'Cell' and 'area' are connectomes. The edges are directed in hierarchies and undirected otherwise. Letters signify the embedders which used this benchmark: I (Boguñá *et al.*, 2010), B (Bläsius *et al.*, 2016), D (Celińska-Kopczyńska and Kopczyński, 2022), P (Nickel and Kiela, 2017), L (Nickel and Kiela, 2018), S (Sala *et al.*, 2018), G (Gu *et al.*, 2019), Y (Yu and De Sa, 2019), M (García-Pérez *et al.*, 2019), T (Sonthalia and Gilbert, 2020), A (Allard and Serrano, 2020; Celinska-Kopczynska and Kopczynski, 2024a), C (Balogh *et al.*, 2025). Small letters appear in the repository but are not discussed in the paper.

## F  ARTIFICIAL NETWORKS

Table 6 and Figure 15 show the details of our evaluation of BFKL versus Lorentz 2D on artificial networks.

## G  DISCREPANCIES

In Table 7, our results are compared to the results obtained in Nickel and Kiela (2017; 2018). Note that VERB is different than VERBF used in our paper, which includes one extra node that is an ancestor of every other node. Furthermore, the ACM hierarchy in Nickel and Kiela (2018) is given as 2299 nodes and 6526 edges, while ours has 2114 nodes and 8121 edges; and the MESH hierarchy is given as 28470 nodes and 191849 edges, while ours has 58737 nodes and 300290 edges.

| | graph name | noun | mammal | verbf | acm | mesh | tetrap | csphd |
|---|---|---|---|---|---|---|---|---|
| | nodes | 82115 | 1180 | 13543 | 2114 | 58737 | 11262 | 1025 |
| | edges | 743086 | 6540 | 48621 | 8121 | 300287 | 527580 | 3978 |
| embedding time [s] | BFKL | 428 | 66 | 37 | 4 | 342 | 738 | 2 |
| | BFKL + DHRG | 1637 | 70 | 600 | 8 | 1191 | 1596 | 4 |
| | Poincare 2D | 51794 | 408 | 2544 | 443 | 19137 | 26256 | 232 |
| | Poincare 2D + DHRG | 52104 | 410 | 2569 | 446 | 19339 | 26295 | 234 |
| | Poincare 3D | 51347 | 369 | 2778 | 457 | 20941 | 26648 | 233 |
| | Lorentz 2D | 38578 | 269 | 2057 | 333 | 19832 | 19706 | 239 |
| | Lorentz 2D + DHRG | 38995 | 277 | 2310 | 336 | 20008 | 20126 | 243 |
| | Lorentz 3D | 39850 | 279 | 2259 | 332 | 15700 | 19892 | 182 |
| | coalescent2 | - | 5 | - | 23 | 67132 | - | 5 |
| | coalescent3 | - | 12 | - | 56 | 14122966 | - | 9 |
| | KVK | - | 1877 | - | 3561 | - | - | 1179 |
| | CLOVE | 117 | 1 | 12 | 2 | 76 | 18 | 1 |
| | CLOVE + DHRG | 3376 | 4 | 282 | 8 | 1162 | 377 | 3 |
| | LPCS | 3 | 1 | 1 | 1 | 1 | 2 | 1 |
| | HMCS | 14 | 1 | 2 | 1 | 11 | 4 | - |
| | Mercator fast | 37202 | 18 | 914 | 38 | 16617 | 1770 | 2 |
| | Mercator full | - | 41 | 4729 | 113 | 104459 | 4802 | 23 |
| | d-Mercator | - | 151 | 4450 | 214 | 65667 | 24761 | 48 |
| | ltiling | - | - | 63074 | 5613 | - | - | 2037 |
| radius [absolute units] | BFKL | 30.992 | 21.120 | 20.733 | 14.945 | 26.835 | 25.632 | 13.639 |
| | BFKL + DHRG | - | - | - | 17.630 | - | - | 15.915 |
| | Poincare 2D | 12.207 | 12.205 | 12.208 | 12.205 | 12.207 | 12.205 | 12.200 |
| | Poincare 3D | 12.208 | 12.205 | 12.202 | 12.118 | 12.207 | 12.205 | 12.199 |
| | Lorentz 2D | 14.509 | 14.509 | 14.509 | 13.290 | 14.509 | 14.509 | 14.509 |
| | Lorentz 2D + DHRG | 17.605 | 16.323 | 16.481 | 14.598 | 16.509 | 17.514 | 16.631 |
| | Lorentz 3D | 14.509 | 13.194 | 13.412 | 11.879 | 12.717 | 14.509 | 13.679 |
| | penalty | 26.008 | 21.386 | 22.354 | 16.511 | 25.933 | 24.706 | 14.443 |
| | anneal2 | - | 7.751 | - | 7.739 | - | - | 7.740 |
| | anneal3 | - | 3.712 | - | 3.712 | | | 3.712 |
| | coalescent2 | - | 14.146 | - | 15.313 | - | - | 13.865 |
| | coalescent3 | - | 14.146 | - | 15.313 | - | - | 13.865 |
| | KVK | - | 15.650 | - | 14.755 | - | - | 14.627 |
| | CLOVE | 22.632 | 14.146 | 19.027 | 15.313 | 21.962 | 18.658 | 13.865 |
| | CLOVE + DHRG | - | 15.382 | - | 16.234 | - | - | 15.864 |
| | LPCS | 22.632 | 14.146 | 19.027 | 15.313 | 21.962 | 18.658 | 13.865 |
| | HMCS | 22.632 | 14.146 | 19.027 | 15.313 | 21.962 | 18.658 | - |
| | HypViewer | 27.523 | 10.703 | 14.963 | 7.145 | 17.330 | 72.251 | 4.114 |
| | Mercator fast | 53.492 | 19.895 | 38.833 | 25.654 | 49.917 | 38.554 | 26.238 |
| | Mercator full | - | 18.450 | 38.899 | 20.254 | 38.062 | 28.056 | 26.055 |
| | d-Mercator | - | 12.647 | 18.262 | 12.152 | 17.665 | 16.871 | 18.676 |
| | ltiling | - | - | 13.285 | 12.508 | - | - | 13.885 |
| MAP: Mean Average Precision | BFKL | 0.284 | 0.219 | 0.348 | 0.423 | 0.321 | 0.276 | 0.667 |
| | BFKL + DHRG | - | - | - | 0.532 | - | - | 0.615 |
| | Poincare 2D | 0.105 | 0.792 | 0.330 | 0.657 | 0.195 | 0.530 | 0.863 |
| | Poincare 3D | 0.492 | 0.943 | 0.526 | 0.852 | 0.376 | 0.917 | 0.903 |
| | Poincare 5D | - | 0.951 | - | - | - | - | 0.844 |
| | Lorentz 2D | 0.194 | 0.835 | 0.246 | 0.600 | 0.183 | 0.699 | 0.845 |
| | Lorentz 2D + DHRG | 0.322 | 0.856 | 0.554 | 0.679 | 0.417 | 0.712 | 0.678 |
| | Lorentz 3D | 0.506 | 0.951 | 0.528 | 0.853 | 0.376 | 0.942 | 0.898 |
| | penalty | 0.330 | 0.299 | 0.294 | 0.410 | 0.293 | 0.689 | 0.691 |
| | anneal2 | - | 0.321 | - | 0.256 | - | - | 0.780 |
| | anneal3 | - | 0.275 | - | 0.240 | | | 0.836 |
| | coalescent2 | - | 0.494 | - | 0.359 | - | - | 0.376 |
| | coalescent3 | - | 0.139 | - | 0.085 | - | - | 0.423 |
| | KVK | - | 0.705 | - | 0.729 | - | - | 0.624 |
| | CLOVE | 0.776 | 0.826 | 0.778 | 0.860 | 0.816 | 0.596 | 0.653 |
| | CLOVE + DHRG | - | 0.865 | - | 0.838 | - | - | 0.612 |
| | LPCS | 0.341 | 0.585 | 0.574 | 0.538 | 0.528 | 0.481 | 0.342 |
| | HMCS | 0.485 | 0.711 | 0.626 | 0.484 | 0.541 | 0.494 | - |
| | HypViewer | 0.047 | 0.124 | 0.134 | 0.134 | 0.122 | 0.014 | 0.416 |
| | Mercator fast | 0.495 | 0.695 | 0.622 | 0.512 | 0.456 | 0.645 | 0.560 |
| | Mercator full | - | 0.842 | 0.727 | 0.753 | 0.548 | 0.753 | 0.672 |
| | d-Mercator | - | 0.054 | 0.023 | 0.033 | 0.009 | 0.014 | 0.571 |
| | ltiling | - | - | 0.201 | 0.522 | - | - | 0.857 |
| | Euclidean 50D | 0.921 | 0.999 | 0.923 | 0.999 | 0.824 | 0.997 | 1.000 |
| | Euclidean 200D | 0.946 | 1.000 | 0.931 | | 0.871 | 0.998 | 1.000 |
| MR: MeanRank | BFKL | 62.6 | 43.1 | 16.7 | 9.3 | 22.3 | 102.1 | 35.8 |
| | BFKL + DHRG | - | - | - | 6.1 | - | - | 45.0 |
| | Poincare 2D | 89.3 | 2.1 | 13.1 | 3.2 | 42.5 | 37.4 | 5.9 |
| | Poincare 3D | 16.0 | 1.2 | 8.5 | 1.7 | 25.3 | 9.7 | 4.1 |
| | Poincare 5D | - | 1.2 | - | - | - | - | 3.9 |
| | Lorentz 2D | 42.3 | 1.8 | 25.7 | 4.0 | 54.5 | 21.5 | 6.5 |
| | Lorentz 2D + DHRG | 30.7 | 1.9 | 4.9 | 2.8 | 14.1 | 31.3 | 20.4 |
| | Lorentz 3D | 15.1 | 1.2 | 8.4 | 1.7 | 25.3 | 7.2 | 4.4 |
| | penalty | 48.2 | 16.2 | 18.2 | 8.3 | 18.9 | 16.9 | 39.7 |
| | anneal2 | - | 12.2 | - | 20.2 | - | - | 25.6 |
| | anneal3 | - | 17.9 | - | 27.4 | - | - | 13.9 |
| | coalescent2 | - | 9.5 | - | 22.4 | - | - | 41.0 |
| | coalescent3 | - | 61.4 | - | 58.2 | - | - | 31.5 |
| | KVK | - | 3.3 | - | 3.0 | - | - | 27.0 |
| | CLOVE | 19.0 | 2.7 | 3.9 | 1.9 | 2.7 | 47.1 | 27.5 |
| | CLOVE + DHRG | - | 2.3 | - | 2.1 | - | - | 28.3 |
| | LPCS | 280.7 | 9.9 | 14.1 | 5.8 | 79.9 | 118.7 | 44.2 |
| | HMCS | 35.7 | 4.1 | 6.8 | 19.0 | 22.6 | 87.8 | - |
| | HypViewer | 4452.4 | 145.7 | 276.1 | 77.0 | 522.2 | 5559.7 | 468.5 |
| | Mercator fast | 177.8 | 8.8 | 16.2 | 20.2 | 89.3 | 106.1 | 29.8 |
| | Mercator full | - | 2.8 | 7.7 | 4.2 | 29.6 | 19.9 | 21.7 |
| | d-Mercator | - | 250.8 | 804.2 | 237.2 | 5150.2 | 5522.4 | 40.3 |
| | ltiling | - | - | 39.5 | 5.4 | - | - | 5.1 |
| | Euclidean 50D | 1.5 | 1.0 | 1.2 | 1.0 | 2.1 | 1.0 | 1.0 |
| | Euclidean 200D | 1.3 | 1.0 | 1.1 | - | 1.6 | 1.0 | 1.0 |

Table 3: Our results on real-world hierarchies. Darker cell color indicate better results.

| | graph name | astrop | condma | grqc | hepph | facebo | yeast | diseas | follow | openfl | ias | CElegs | Human1 | Droso1 | Mouse3 |
|---|---|---|---|---|---|---|---|---|---|---|---|---|---|---|---|
| | nodes | 17903 | 21363 | 4158 | 11204 | 4039 | 1458 | 516 | 74946 | 3397 | 23748 | 279 | 493 | 350 | 1076 |
| | edges (undirected) | 196972 | 91286 | 13422 | 117619 | 88234 | 1948 | 1188 | 537952 | 19230 | 58414 | 2287 | 7773 | 2887 | 90811 |
| **embedding time [s]** | BFKL | 179 | 91 | 7 | 82 | 26 | 1 | 1 | 269 | 11 | 40 | 1 | 2 | 1 | 17 |
| | BFKL + DHRG | 550 | 579 | 79 | 168 | 43 | 26 | 2 | 2634 | 29 | 440 | 3 | 5 | 2 | 45 |
| | Poincare 2D | 20481 | 10025 | 1321 | 11920 | 8580 | 235 | 155 | 65600 | 2020 | 7396 | 254 | 807 | 323 | 8781 |
| | Poincare 2D + DHRG | 20687 | 10265 | 1335 | 11993 | 8618 | 240 | 157 | 65878 | 2030 | 7476 | 255 | 815 | 324 | 8792 |
| | Poincare 3D | 20485 | 10002 | 1366 | 12153 | 8624 | 235 | 156 | 68445 | 2061 | 7902 | 260 | 795 | 328 | 8716 |
| | Lorentz 2D | 14741 | 7349 | 986 | 8761 | 6317 | 183 | 130 | 49531 | 1501 | 5604 | 204 | 639 | 255 | 6413 |
| | Lorentz 2D + DHRG | 14891 | 7475 | 1005 | 8807 | 6371 | 200 | 137 | 49814 | 1517 | 5691 | 208 | 642 | 256 | 6436 |
| | Lorentz 3D | 14612 | 7479 | 1155 | 9392 | 6319 | 184 | 134 | 51675 | 1485 | 5778 | 216 | 1554 | 248 | 6252 |
| | coalescent2 | 13557 | 23340 | 162 | 3147 | 164 | 9 | 3 | - | 83 | - | 2 | 4 | 3 | 35 |
| | coalescent3 | - | - | 373 | 15012 | 499 | 20 | 4 | - | 259 | - | 2 | 5 | 3 | 40 |
| | KVK | - | - | 13910 | - | 15250 | 1278 | 205 | - | 7918 | - | 99 | 1 | 156 | 1 |
| | CLOVE | 26 | 31 | 5 | 15 | 7 | 2 | 1 | 188 | 4 | 17 | 1 | 1 | 1 | 3 |
| | CLOVE + DHRG | 279 | 321 | 55 | 339 | 161 | 4 | 2 | 1560 | 59 | 405 | 2 | 6 | 3 | 134 |
| | LPCS | 1 | 3 | 1 | 2 | 1 | 1 | 1 | 5 | 1 | 1 | 1 | 1 | 1 | 1 |
| | HMCS | 6 | 5 | 1 | 3 | 2 | 1 | 1 | 17 | 1 | 4 | 1 | 1 | 1 | 1 |
| | Mercator fast | 1565 | 2088 | 85 | 587 | 33 | 4 | 4 | 29707 | 56 | 2721 | 5 | 6 | 11 | 39 |
| | Mercator full | 8906 | 15167 | 454 | 3797 | 408 | 47 | 7 | 195644 | 338 | 18053 | 7 | 9 | 13 | 65 |
| | d-Mercator | 5110 | 7393 | 313 | 2291 | 403 | 44 | 27 | - | 259 | 9136 | 21 | 38 | 24 | 528 |
| | ltiling | - | - | 19721 | - | - | 3200 | 1592 | - | 25772 | - | 2795 | 7307 | - | 7237 |
| **radius [absolute units]** | BFKL | 15.430 | 17.677 | 21.792 | 21.883 | 12.576 | 16.267 | 12.680 | 20.904 | 25.030 | 23.228 | 7.787 | 7.421 | 8.178 | 8.625 |
| | BFKL + DHRG | - | - | - | - | - | 14.906 | 18.801 | 14.146 | - | - | 9.250 | 8.747 | 9.796 | 10.060 |
| | Poincare 2D | 12.199 | 12.197 | 11.666 | 12.209 | 12.199 | 11.455 | 12.198 | 12.146 | 12.205 | 12.206 | 6.702 | 12.196 | 8.205 | 10.793 |
| | Poincare 3D | 12.207 | 10.906 | 10.493 | 12.199 | 12.200 | 9.571 | 10.336 | 11.123 | - | 12.199 | 9.240 | 9.762 | 11.025 | 12.199 |
| | Lorentz 2D | - | 10.941 | 10.962 | 12.335 | 11.555 | 9.563 | 11.142 | 10.698 | 12.261 | 13.250 | 6.483 | 11.664 | 7.713 | 10.492 |
| | Lorentz 2D + DHRG | 13.552 | 12.967 | 12.250 | 14.099 | 13.181 | 11.228 | 12.301 | 12.425 | 13.640 | 15.312 | 7.588 | 11.384 | 8.267 | 11.665 |
| | Lorentz 3D | 12.415 | 11.088 | 10.178 | 12.649 | 12.930 | 9.707 | 10.369 | 10.671 | 12.323 | 12.589 | 9.541 | 9.349 | 10.523 | 12.521 |
| | penalty | 14.229 | 13.978 | 14.613 | 15.823 | 10.267 | 15.077 | 8.448 | 16.588 | 23.986 | 24.409 | 8.038 | 7.713 | 8.015 | 10.060 |
| | anneal2 | - | - | 7.784 | - | 7.782 | 7.775 | 7.774 | - | 7.763 | - | 7.690 | 7.602 | 7.680 | 7.751 |
| | anneal3 | - | - | 3.735 | - | 3.747 | 3.694 | 3.725 | - | 3.712 | - | 3.650 | 3.692 | 3.650 | 3.702 |
| | coalescent2 | 19.585 | 19.939 | 16.666 | 18.648 | 16.608 | 14.570 | 12.492 | - | 16.261 | - | 11.262 | 12.401 | 11.716 | 13.962 |
| | coalescent3 | - | - | 16.666 | 18.648 | 16.608 | 14.570 | 12.492 | - | 16.261 | - | 11.262 | 12.401 | 11.716 | 13.962 |
| | KVK | - | - | 18.332 | - | 17.367 | 15.119 | 12.972 | - | - | - | 10.063 | - | 11.699 | - |
| | CLOVE | 19.585 | 19.939 | 16.666 | 18.648 | 16.608 | 14.570 | 12.492 | 22.449 | 16.261 | 20.151 | 11.262 | 12.401 | 11.716 | 13.962 |
| | CLOVE + DHRG | - | - | - | - | - | 16.208 | 14.384 | - | - | - | 13.283 | 13.835 | 13.753 | 15.710 |
| | LPCS | 19.585 | 19.939 | 16.666 | 18.648 | 16.608 | 14.570 | 12.492 | 22.449 | 16.261 | 20.151 | 11.262 | 12.401 | 11.716 | 13.962 |
| | HMCS | 19.585 | 19.939 | 16.666 | 18.648 | 16.608 | - | 12.492 | 22.449 | 16.261 | 20.151 | 11.262 | 12.401 | 11.716 | 13.962 |
| | Mercator fast | 70.081 | 64.709 | 42.147 | 53.510 | 31.315 | 22.056 | 26.306 | 173.262 | 33.530 | 63.496 | 16.438 | 20.775 | 23.148 | 28.747 |
| | Mercator full | 56.490 | 52.331 | 40.502 | 51.965 | 30.507 | 25.973 | 24.869 | - | 35.790 | 63.223 | 16.261 | 15.998 | 24.585 | 30.338 |
| | d-Mercator | 12.119 | 14.326 | 22.544 | 19.129 | 17.288 | 11.746 | 10.418 | - | 10.304 | 17.880 | 13.956 | 14.604 | 16.448 | 16.730 |
| | ltiling | - | - | 11.157 | - | - | 8.822 | 9.663 | - | 12.055 | - | 6.485 | 9.193 | - | - |
| **MAP: Mean Average Precision** | BFKL | 0.208 | 0.278 | 0.480 | 0.320 | 0.531 | 0.756 | 0.827 | 0.128 | 0.459 | 0.547 | 0.454 | 0.575 | 0.381 | 0.558 |
| | BFKL + DHRG | - | - | - | - | 0.541 | 0.755 | 0.829 | - | - | - | 0.458 | 0.583 | 0.387 | 0.575 |
| | Poincare 2D | 0.321 | 0.392 | 0.642 | 0.471 | 0.603 | 0.685 | 0.889 | 0.048 | 0.438 | 0.193 | 0.492 | 0.630 | 0.384 | 0.576 |
| | Poincare 3D | 0.462 | 0.597 | 0.781 | 0.578 | 0.683 | 0.772 | 0.929 | 0.135 | - | 0.472 | 0.576 | 0.722 | 0.480 | 0.652 |
| | Poincare 5D | 0.512 | 0.670 | 0.828 | 0.629 | 0.714 | 0.872 | 0.933 | 0.187 | 0.548 | 0.666 | 0.600 | 0.728 | 0.519 | 0.671 |
| | Lorentz 2D | - | 0.375 | 0.614 | 0.456 | 0.607 | 0.542 | 0.889 | 0.046 | 0.413 | 0.148 | 0.494 | 0.654 | 0.386 | 0.574 |
| | Lorentz 2D + DHRG | 0.439 | 0.565 | 0.745 | 0.530 | 0.632 | 0.796 | 0.897 | 0.098 | 0.475 | 0.355 | 0.482 | 0.651 | 0.369 | 0.586 |
| | Lorentz 3D | 0.464 | 0.597 | 0.785 | 0.576 | 0.683 | 0.758 | 0.920 | 0.129 | 0.509 | 0.471 | 0.577 | 0.722 | 0.495 | 0.651 |
| | penalty | 0.238 | 0.211 | 0.461 | 0.365 | 0.540 | 0.755 | 0.779 | 0.111 | 0.442 | 0.469 | 0.441 | 0.588 | 0.375 | 0.573 |
| | anneal2 | - | - | 0.613 | - | 0.603 | 0.653 | 0.841 | - | 0.393 | - | 0.535 | 0.675 | 0.479 | 0.611 |
| | anneal3 | - | - | 0.649 | - | 0.626 | 0.672 | 0.914 | - | 0.421 | - | 0.589 | 0.783 | 0.517 | 0.654 |
| | coalescent2 | 0.084 | 0.015 | 0.058 | 0.106 | 0.365 | 0.387 | 0.585 | - | 0.208 | - | 0.302 | 0.453 | 0.222 | 0.506 |
| | coalescent3 | - | - | 0.224 | 0.185 | 0.215 | 0.359 | 0.565 | - | 0.161 | - | 0.303 | 0.528 | 0.289 | 0.491 |
| | KVK | - | - | 0.657 | - | 0.587 | 0.760 | 0.814 | - | - | - | 0.496 | - | 0.418 | - |
| | CLOVE | 0.512 | 0.625 | 0.685 | 0.584 | 0.555 | 0.777 | 0.807 | 0.437 | 0.580 | 0.698 | 0.461 | 0.575 | 0.415 | 0.463 |
| | CLOVE + DHRG | - | - | - | - | - | 0.787 | 0.813 | - | - | - | 0.467 | 0.580 | 0.418 | 0.552 |
| | LPCS | 0.302 | 0.464 | 0.556 | 0.373 | 0.266 | 0.645 | 0.624 | 0.148 | 0.220 | 0.565 | 0.276 | 0.339 | 0.216 | 0.408 |
| | HMCS | 0.469 | 0.567 | 0.613 | 0.524 | 0.519 | - | 0.673 | 0.372 | 0.517 | 0.581 | 0.443 | 0.567 | 0.384 | 0.539 |
| | Mercator fast | 0.172 | 0.222 | 0.387 | 0.253 | 0.368 | 0.680 | 0.805 | 0.218 | 0.488 | 0.455 | 0.336 | 0.411 | 0.270 | 0.520 |
| | Mercator full | 0.244 | 0.265 | 0.467 | 0.331 | 0.494 | 0.754 | 0.861 | - | 0.557 | 0.523 | 0.484 | 0.552 | 0.435 | 0.584 |
| | d-Mercator | 0.044 | 0.078 | 0.319 | 0.193 | 0.310 | 0.216 | 0.395 | - | 0.113 | 0.045 | 0.370 | 0.635 | 0.339 | 0.583 |
| | orig TreeRep rec | - | 0.437 | 0.680 | 0.515 | 0.355 | 0.817 | 0.894 | - | 0.572 | - | 0.205 | 0.241 | 0.243 | 0.233 |
| | orig TreeRep norec | - | 0.492 | 0.668 | 0.512 | 0.360 | 0.816 | 0.852 | - | 0.561 | - | 0.204 | 0.259 | 0.250 | 0.227 |
| | ltiling | - | - | 0.589 | - | - | 0.519 | 0.877 | - | 0.402 | - | 0.488 | 0.637 | - | - |
| | Euclidean 50D | 0.988 | 0.968 | 1.000 | 0.980 | 1.000 | 1.000 | 1.000 | 0.452 | 1.000 | 0.831 | 1.000 | 1.000 | 1.000 | 0.943 |
| | Euclidean 200D | 0.994 | 0.975 | 1.000 | 0.984 | 1.000 | 1.000 | 1.000 | 0.826 | 1.000 | 0.866 | 1.000 | 1.000 | 1.000 | 1.000 |
| **MR: MeanRank** | BFKL | 1880.0 | 1717.0 | 169.2 | 558.2 | 84.0 | 50.4 | 9.2 | 7660.1 | 56.1 | 709.7 | 38.0 | 50.9 | 52.0 | 104.0 |
| | BFKL + DHRG | - | - | - | - | 82.1 | 45.9 | 8.7 | - | - | - | 37.6 | 48.6 | 49.7 | 99.7 |
| | Poincare 2D | 1153.7 | 887.3 | 76.0 | 298.5 | 45.3 | 32.7 | 6.5 | 6567.3 | 67.8 | 537.2 | 31.6 | 51.4 | 46.6 | 96.2 |
| | Poincare 3D | 924.3 | 647.8 | 49.8 | 242.3 | 43.0 | 21.2 | 3.8 | 5522.7 | - | 359.7 | 28.1 | 25.1 | 39.3 | 84.1 |
| | Poincare 5D | 725.7 | 491.9 | 36.5 | 206.4 | 32.6 | 13.7 | 3.4 | 4894.0 | 51.5 | 281.4 | 25.0 | 24.7 | 35.0 | 80.0 |
| | Lorentz 2D | - | 949.1 | 81.4 | 293.2 | 55.2 | 37.7 | 6.7 | 6708.6 | 70.4 | 560.5 | 31.5 | 44.5 | 46.9 | 96.8 |
| | Lorentz 2D + DHRG | 1160.8 | 1069.2 | 89.8 | 320.8 | 57.3 | 33.2 | 7.5 | 6688.8 | 78.4 | 677.2 | 32.8 | 44.3 | 47.6 | 96.5 |
| | Lorentz 3D | 876.1 | 647.1 | 51.5 | 242.6 | 44.9 | 22.2 | 4.3 | 5585.8 | 58.0 | 337.7 | 27.2 | 25.0 | 38.7 | 84.2 |
| | penalty | 1571.0 | 1496.5 | 144.6 | 481.6 | 62.9 | 52.0 | 8.5 | 6989.7 | 65.3 | 911.0 | 37.0 | 46.2 | 49.1 | 97.9 |
| | anneal2 | - | - | 163.1 | - | 71.8 | 69.9 | 16.1 | - | 70.3 | - | 33.4 | 40.4 | 46.3 | 94.0 |
| | anneal3 | - | - | 116.5 | - | 38.6 | 63.9 | 7.8 | - | 52.4 | - | 27.3 | 23.3 | 37.8 | 79.7 |
| | coalescent2 | 2086.1 | 3404.7 | 437.4 | 623.6 | 96.3 | 63.2 | 14.4 | - | 115.6 | - | 38.6 | 41.1 | 59.1 | 107.8 |
| | coalescent3 | - | - | 186.1 | 512.6 | 218.8 | 45.4 | 11.6 | - | 164.9 | - | 41.5 | 28.7 | 49.8 | 113.8 |
| | KVK | - | - | 128.6 | - | 79.2 | 47.4 | 9.3 | - | - | - | 34.6 | - | 47.6 | - |
| | CLOVE | 1821.7 | 1479.9 | 159.8 | 750.8 | 80.6 | 48.2 | 12.2 | 6879.8 | 111.1 | 684.0 | 43.4 | 52.7 | 62.1 | 166.0 |
| | CLOVE + DHRG | - | - | - | - | - | 42.2 | 11.2 | - | - | - | 41.8 | 51.2 | 52.6 | 120.5 |
| | LPCS | 1855.9 | 1359.6 | 116.5 | 398.2 | 140.6 | 46.3 | 11.8 | 7656.8 | 158.9 | 686.1 | 50.3 | 64.6 | 73.8 | 149.9 |
| | HMCS | 1827.8 | 1355.3 | 120.6 | 420.3 | 91.7 | - | 13.3 | 7563.1 | 90.8 | 670.6 | 43.9 | 54.6 | 66.9 | 127.1 |
| | Mercator fast | 2081.3 | 2046.1 | 209.5 | 686.9 | 101.4 | 51.0 | 7.7 | 8073.4 | 69.0 | 951.3 | 37.7 | 41.5 | 54.4 | 103.5 |
| | Mercator full | 1513.3 | 1539.9 | 153.9 | 476.8 | 77.8 | 45.7 | 6.7 | - | 61.5 | 770.3 | 34.2 | 41.2 | 47.5 | 99.7 |
| | d-Mercator | 5590.7 | 4547.3 | 505.4 | 1659.3 | 106.7 | 172.5 | 27.3 | - | 430.9 | 4560.5 | 34.3 | 24.1 | 41.2 | 92.8 |
| | ltiling | - | - | 80.7 | - | - | 39.8 | 5.6 | - | 69.5 | - | 31.5 | 43.0 | - | - |
| | Euclidean 50D | 1.0 | 1.0 | 1.0 | 1.0 | 1.0 | 1.0 | 1.0 | 117.6 | 1.0 | 1.2 | 1.0 | 1.0 | 1.0 | 15.3 |
| | Euclidean 200D | 1.0 | 1.0 | 1.0 | 1.0 | 1.0 | 1.0 | 1.0 | 1.1 | 1.0 | 1.1 | 1.0 | 1.0 | 1.0 | 1.0 |

Table 4: Our results on real-world networks. Darker cell color indicate better results.

| | graph name | astrop | condma | grqc | hepph | facebo | yeast | diseas | follow | openfl | ias | CElegs | Human1 | Droso1 | Mouse3 |
|---|---|---|---|---|---|---|---|---|---|---|---|---|---|---|---|
| **GSR: Greedy Success Ratio** | BFKL | 0.060 | 0.026 | 0.052 | 0.072 | 0.463 | 0.061 | 0.153 | 0.047 | 0.545 | 0.494 | 0.775 | 0.778 | 0.649 | 0.904 |
| | BFKL + DHRG | - | - | - | - | 0.451 | 0.064 | 0.186 | - | - | - | 0.776 | 0.809 | 0.652 | 0.919 |
| | Poincare 2D | 0.142 | 0.060 | 0.101 | 0.153 | 0.515 | 0.149 | 0.233 | 0.036 | 0.392 | 0.409 | 0.897 | 0.841 | 0.727 | 0.943 |
| | Poincare 3D | 0.259 | 0.146 | 0.176 | 0.200 | 0.542 | 0.216 | 0.261 | 0.098 | - | 0.697 | 0.933 | 0.915 | 0.847 | 0.964 |
| | Lorentz 2D | - | 0.053 | 0.099 | 0.169 | 0.460 | 0.141 | 0.220 | 0.035 | 0.395 | 0.360 | 0.899 | 0.880 | 0.747 | 0.943 |
| | Lorentz 2D + DHRG | 0.227 | 0.103 | 0.115 | 0.173 | 0.437 | 0.135 | 0.175 | 0.037 | 0.402 | 0.506 | 0.840 | 0.923 | 0.685 | 0.923 |
| | Lorentz 3D | 0.272 | 0.144 | 0.176 | 0.208 | 0.474 | 0.249 | 0.270 | 0.096 | 0.455 | 0.712 | 0.929 | 0.921 | 0.841 | 0.962 |
| | penalty | 0.073 | 0.017 | 0.054 | 0.085 | 0.384 | 0.062 | 0.230 | 0.026 | 0.517 | 0.477 | 0.775 | 0.757 | 0.636 | 0.915 |
| | anneal2 | - | - | 0.072 | - | 0.398 | 0.086 | 0.152 | - | 0.338 | - | 0.929 | 0.942 | 0.855 | 0.965 |
| | anneal3 | - | - | 0.102 | - | 0.515 | 0.117 | 0.223 | - | 0.390 | - | 0.952 | 0.896 | 0.847 | 0.950 |
| | coalescent2 | 0.034 | 0.006 | 0.019 | 0.055 | 0.371 | 0.045 | 0.128 | - | 0.249 | - | 0.465 | 0.529 | 0.427 | 0.808 |
| | coalescent3 | - | - | 0.055 | 0.072 | 0.300 | 0.051 | 0.160 | - | 0.226 | - | 0.625 | 0.616 | 0.569 | 0.831 |
| | KVK | - | - | 0.109 | - | 0.413 | 0.098 | 0.164 | - | - | - | 0.892 | - | 0.779 | - |
| | CLOVE | 0.434 | 0.242 | 0.140 | 0.308 | 0.471 | 0.111 | 0.166 | 0.506 | 0.662 | 0.784 | 0.866 | 0.862 | 0.844 | 0.972 |
| | CLOVE + DHRG | - | - | - | - | - | 0.101 | 0.160 | - | - | - | 0.823 | 0.801 | 0.780 | 0.934 |
| | LPCS | 0.152 | 0.140 | 0.084 | 0.119 | 0.327 | 0.084 | 0.128 | 0.107 | 0.189 | 0.631 | 0.536 | 0.471 | 0.433 | 0.780 |
| | HMCS | 0.367 | 0.194 | 0.105 | 0.193 | 0.471 | - | 0.156 | 0.441 | 0.483 | 0.680 | 0.820 | 0.828 | 0.793 | 0.943 |
| | Mercator fast | 0.031 | 0.014 | 0.038 | 0.053 | 0.365 | 0.048 | 0.153 | 0.062 | 0.437 | 0.430 | 0.524 | 0.534 | 0.437 | 0.829 |
| | Mercator full | 0.140 | 0.043 | 0.068 | 0.122 | 0.442 | 0.068 | 0.195 | - | 0.568 | 0.526 | 0.868 | 0.784 | 0.783 | 0.960 |
| | d-Mercator | 0.014 | 0.005 | 0.013 | 0.029 | 0.127 | 0.023 | 0.040 | - | 0.127 | 0.066 | 0.587 | 0.786 | 0.562 | 0.880 |
| | ltiling | - | - | 0.102 | - | - | 0.126 | 0.205 | - | 0.369 | - | 0.897 | 0.855 | - | - |
| **GRE: Greedy Routing Efficiency** | BFKL | 0.055 | 0.025 | 0.050 | 0.068 | 0.451 | 0.060 | 0.149 | 0.044 | 0.526 | 0.482 | 0.688 | 0.671 | 0.574 | 0.848 |
| | BFKL + DHRG | - | - | - | - | 0.442 | 0.063 | 0.179 | - | - | - | 0.691 | 0.696 | 0.582 | 0.854 |
| | Poincare 2D | 0.120 | 0.054 | 0.094 | 0.137 | 0.496 | 0.142 | 0.228 | 0.032 | 0.370 | 0.397 | 0.778 | 0.733 | 0.630 | 0.870 |
| | Poincare 3D | 0.221 | 0.127 | 0.158 | 0.177 | 0.534 | 0.202 | 0.253 | 0.087 | - | 0.666 | 0.830 | 0.818 | 0.758 | 0.933 |
| | Lorentz 2D | - | 0.047 | 0.092 | 0.150 | 0.442 | 0.134 | 0.213 | 0.030 | 0.372 | 0.348 | 0.778 | 0.769 | 0.644 | 0.867 |
| | Lorentz 2D + DHRG | 0.191 | 0.090 | 0.106 | 0.154 | 0.428 | 0.130 | 0.172 | 0.033 | 0.378 | 0.487 | 0.736 | 0.799 | 0.597 | 0.853 |
| | Lorentz 3D | 0.231 | 0.126 | 0.159 | 0.183 | 0.468 | 0.234 | 0.261 | 0.085 | 0.435 | 0.681 | 0.832 | 0.822 | 0.756 | 0.931 |
| | penalty | 0.066 | 0.016 | 0.052 | 0.078 | 0.361 | 0.061 | 0.221 | 0.025 | 0.486 | 0.465 | 0.681 | 0.663 | 0.559 | 0.841 |
| | anneal2 | - | - | 0.067 | - | 0.382 | 0.082 | 0.149 | - | 0.313 | - | 0.812 | 0.814 | 0.745 | 0.893 |
| | anneal3 | - | - | 0.093 | - | 0.488 | 0.111 | 0.219 | - | 0.362 | - | 0.845 | 0.812 | 0.751 | 0.893 |
| | coalescent2 | 0.032 | 0.006 | 0.018 | 0.052 | 0.370 | 0.044 | 0.125 | - | 0.235 | - | 0.432 | 0.481 | 0.395 | 0.776 |
| | coalescent3 | - | - | 0.053 | 0.069 | 0.298 | 0.050 | 0.157 | - | 0.216 | - | 0.584 | 0.553 | 0.529 | 0.805 |
| | KVK | - | - | 0.099 | - | 0.406 | 0.094 | 0.161 | - | - | - | 0.782 | - | 0.684 | - |
| | CLOVE | 0.340 | 0.194 | 0.122 | 0.240 | 0.454 | 0.106 | 0.162 | 0.419 | 0.622 | 0.747 | 0.749 | 0.727 | 0.740 | 0.922 |
| | CLOVE + DHRG | - | - | - | - | - | 0.097 | 0.157 | - | - | - | 0.724 | 0.682 | 0.698 | 0.900 |
| | LPCS | 0.126 | 0.117 | 0.078 | 0.107 | 0.319 | 0.081 | 0.126 | 0.095 | 0.180 | 0.607 | 0.488 | 0.416 | 0.394 | 0.752 |
| | HMCS | 0.294 | 0.161 | 0.097 | 0.171 | 0.447 | - | 0.152 | 0.367 | 0.455 | 0.649 | 0.721 | 0.701 | 0.689 | 0.905 |
| | Mercator fast | 0.030 | 0.014 | 0.037 | 0.050 | 0.363 | 0.047 | 0.150 | 0.058 | 0.421 | 0.424 | 0.480 | 0.476 | 0.407 | 0.785 |
| | Mercator full | 0.121 | 0.040 | 0.063 | 0.110 | 0.434 | 0.067 | 0.190 | - | 0.541 | 0.514 | 0.761 | 0.683 | 0.698 | 0.915 |
| | d-Mercator | 0.013 | 0.005 | 0.013 | 0.027 | 0.113 | 0.023 | 0.039 | - | 0.117 | 0.064 | 0.538 | 0.702 | 0.520 | 0.839 |
| | ltiling | - | - | 0.094 | - | - | 0.120 | 0.201 | - | 0.348 | - | 0.775 | 0.743 | - | - |
| **LL: normalized loglikelihood** | BFKL | 0.254 | 0.271 | 0.487 | 0.452 | 0.554 | 0.586 | 0.651 | 0.230 | 0.457 | 0.579 | 0.283 | 0.371 | 0.259 | 0.348 |
| | BFKL + DHRG | - | - | - | - | 0.605 | 0.608 | 0.693 | - | - | - | 0.302 | 0.397 | 0.293 | 0.401 |
| | Poincare 2D | 0.413 | 0.456 | 0.675 | 0.649 | 0.714 | 0.614 | 0.807 | 0.272 | 0.621 | 0.510 | 0.348 | 0.438 | 0.309 | 0.409 |
| | Poincare 3D | 0.548 | 0.594 | 0.770 | 0.737 | 0.748 | 0.658 | 0.856 | 0.384 | - | 0.633 | 0.415 | 0.552 | 0.388 | 0.461 |
| | Lorentz 2D | - | 0.445 | 0.662 | 0.638 | 0.708 | 0.548 | 0.794 | 0.265 | 0.606 | 0.479 | 0.349 | 0.463 | 0.311 | 0.407 |
| | Lorentz 2D + DHRG | 0.501 | 0.542 | 0.727 | 0.707 | 0.720 | 0.660 | 0.799 | 0.314 | 0.635 | 0.590 | 0.352 | 0.460 | 0.318 | 0.417 |
| | Lorentz 3D | 0.551 | 0.593 | 0.769 | 0.735 | 0.746 | 0.658 | 0.846 | 0.381 | 0.669 | 0.634 | 0.422 | 0.553 | 0.391 | 0.461 |
| | penalty | 0.346 | 0.315 | 0.543 | 0.566 | 0.673 | 0.588 | 0.701 | 0.285 | 0.495 | 0.555 | 0.311 | 0.413 | 0.303 | 0.407 |
| | anneal2 | - | - | 0.634 | - | 0.703 | 0.557 | 0.718 | - | 0.598 | - | 0.367 | 0.475 | 0.344 | 0.426 |
| | anneal3 | - | - | 0.671 | - | 0.726 | 0.565 | 0.812 | - | 0.619 | - | 0.415 | 0.597 | 0.385 | 0.473 |
| | coalescent2 | 0.210 | 0.108 | 0.251 | 0.470 | 0.518 | 0.413 | 0.581 | - | 0.501 | - | 0.222 | 0.329 | 0.181 | 0.348 |
| | coalescent3 | - | - | 0.394 | 0.467 | 0.334 | 0.181 | 0.425 | - | 0.399 | - | 0.226 | 0.386 | 0.242 | 0.332 |
| | KVK | - | - | 0.669 | - | 0.662 | 0.642 | 0.741 | - | - | - | 0.348 | - | 0.327 | - |
| | CLOVE | 0.419 | 0.532 | 0.606 | 0.444 | 0.567 | 0.640 | 0.694 | 0.383 | 0.507 | 0.673 | 0.284 | 0.358 | 0.240 | 0.180 |
| | CLOVE + DHRG | - | - | - | - | - | 0.659 | 0.719 | - | - | - | 0.303 | 0.380 | 0.302 | 0.364 |
| | LPCS | 0.336 | 0.500 | 0.588 | 0.586 | 0.357 | 0.594 | 0.573 | 0.274 | 0.461 | 0.641 | 0.168 | 0.205 | 0.123 | 0.250 |
| | HMCS | 0.373 | 0.519 | 0.625 | 0.606 | 0.488 | - | 0.587 | 0.347 | 0.578 | 0.639 | 0.266 | 0.354 | 0.188 | 0.322 |
| | Mercator fast | 0.253 | 0.246 | 0.425 | 0.437 | 0.545 | 0.535 | 0.696 | 0.250 | 0.587 | 0.482 | 0.262 | 0.308 | 0.251 | 0.368 |
| | Mercator full | 0.384 | 0.361 | 0.554 | 0.574 | 0.624 | 0.608 | 0.753 | - | 0.636 | 0.603 | 0.340 | 0.391 | 0.327 | 0.401 |
| | d-Mercator | 0.022 | 0.037 | 0.254 | 0.185 | 0.425 | 0.174 | 0.334 | - | 0.122 | 0.068 | 0.290 | 0.490 | 0.313 | 0.414 |
| **ICV: Information Control Value** | BFKL | 0.521 | 0.466 | 0.431 | 0.550 | 0.644 | 0.353 | 0.456 | 0.487 | 0.467 | 0.428 | 0.513 | 0.567 | 0.510 | 0.590 |
| | BFKL + DHRG | - | - | - | - | 0.662 | 0.335 | 0.450 | - | - | - | 0.507 | 0.571 | 0.510 | 0.609 |
| | Poincare 2D | 0.587 | 0.552 | 0.575 | 0.666 | 0.723 | 0.433 | 0.513 | 0.538 | 0.598 | 0.513 | 0.539 | 0.573 | 0.533 | 0.613 |
| | Poincare 3D | 0.595 | 0.537 | 0.531 | 0.656 | 0.696 | 0.373 | 0.440 | 0.535 | - | 0.451 | 0.482 | 0.572 | 0.481 | 0.612 |
| | Lorentz 2D | - | 0.553 | 0.578 | 0.662 | 0.723 | 0.450 | 0.529 | 0.538 | 0.597 | 0.501 | 0.540 | 0.588 | 0.533 | 0.613 |
| | Lorentz 2D + DHRG | 0.607 | 0.565 | 0.584 | 0.684 | 0.723 | 0.436 | 0.520 | 0.543 | 0.593 | 0.499 | 0.534 | 0.588 | 0.531 | 0.615 |
| | Lorentz 3D | 0.598 | 0.537 | 0.533 | 0.655 | 0.697 | 0.373 | 0.439 | 0.536 | 0.544 | 0.450 | 0.484 | 0.579 | 0.488 | 0.616 |
| | penalty | 0.559 | 0.519 | 0.537 | 0.625 | 0.709 | 0.368 | 0.535 | 0.521 | 0.497 | 0.459 | 0.523 | 0.586 | 0.526 | 0.612 |
| | anneal2 | - | - | 0.603 | - | 0.730 | 0.495 | 0.558 | - | 0.620 | - | 0.535 | 0.600 | 0.534 | 0.620 |
| | anneal3 | - | - | 0.595 | - | 0.733 | 0.479 | 0.555 | - | 0.614 | - | 0.541 | 0.635 | 0.537 | 0.634 |
| | coalescent2 | 0.496 | 0.425 | 0.427 | 0.567 | 0.616 | 0.353 | 0.441 | - | 0.532 | - | 0.473 | 0.531 | 0.467 | 0.584 |
| | coalescent3 | - | - | 0.366 | 0.506 | 0.523 | 0.288 | 0.348 | - | 0.436 | - | 0.408 | 0.487 | 0.412 | 0.563 |
| | KVK | - | - | 0.493 | - | 0.683 | 0.368 | 0.461 | - | - | - | 0.519 | - | 0.512 | - |
| | CLOVE | 0.553 | 0.510 | 0.507 | 0.561 | - | 0.375 | 0.457 | 0.517 | 0.525 | 0.469 | 0.485 | 0.541 | 0.478 | 0.530 |
| | CLOVE + DHRG | - | - | - | - | - | 0.353 | 0.438 | - | - | - | 0.478 | 0.537 | 0.482 | 0.587 |
| | LPCS | 0.530 | 0.501 | 0.488 | 0.606 | 0.561 | 0.376 | 0.448 | 0.490 | 0.520 | 0.464 | 0.465 | 0.504 | 0.464 | 0.551 |
| | HMCS | 0.541 | 0.506 | 0.496 | 0.612 | 0.605 | - | 0.448 | 0.508 | 0.551 | 0.465 | 0.482 | 0.540 | 0.476 | 0.573 |
| | Mercator fast | 0.435 | 0.344 | 0.349 | 0.482 | 0.601 | 0.290 | 0.348 | 0.299 | 0.467 | 0.290 | 0.459 | 0.506 | 0.434 | 0.579 |
| | Mercator full | 0.479 | 0.380 | 0.370 | 0.520 | 0.632 | 0.274 | 0.362 | - | 0.471 | 0.300 | 0.475 | 0.541 | 0.442 | 0.589 |
| | d-Mercator | 0.480 | 0.452 | 0.386 | 0.485 | 0.575 | 0.353 | 0.414 | - | 0.474 | 0.412 | 0.419 | 0.525 | 0.412 | 0.589 |

Table 5: Our results on real-world networks. Darker cell color indicate better results. Log-likelihood (LL) is normalized by dividing by the value of $N$ from Section C and subtracting from 1.

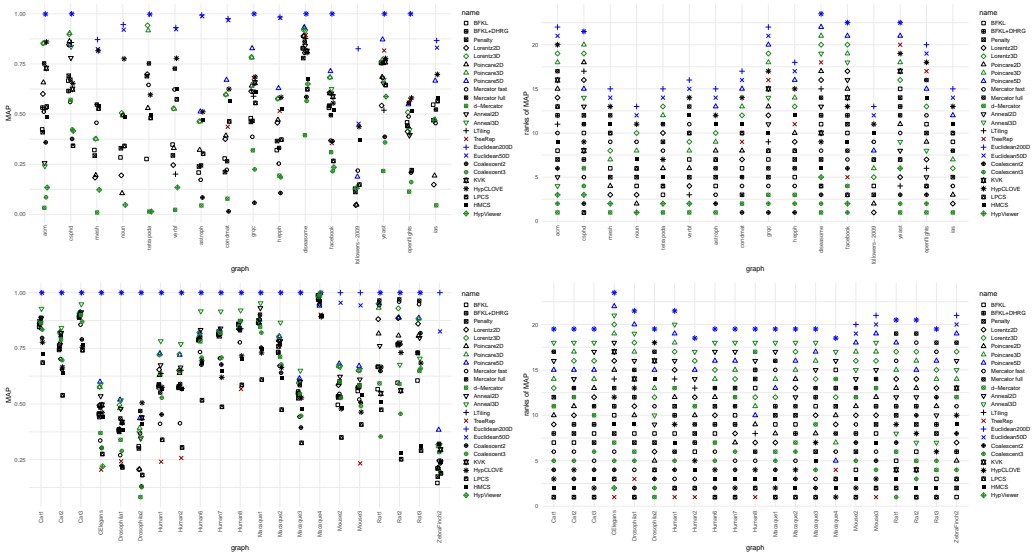

Figure 7: Quality assessment of embedders on real-world hierarchies and networks: MAP.

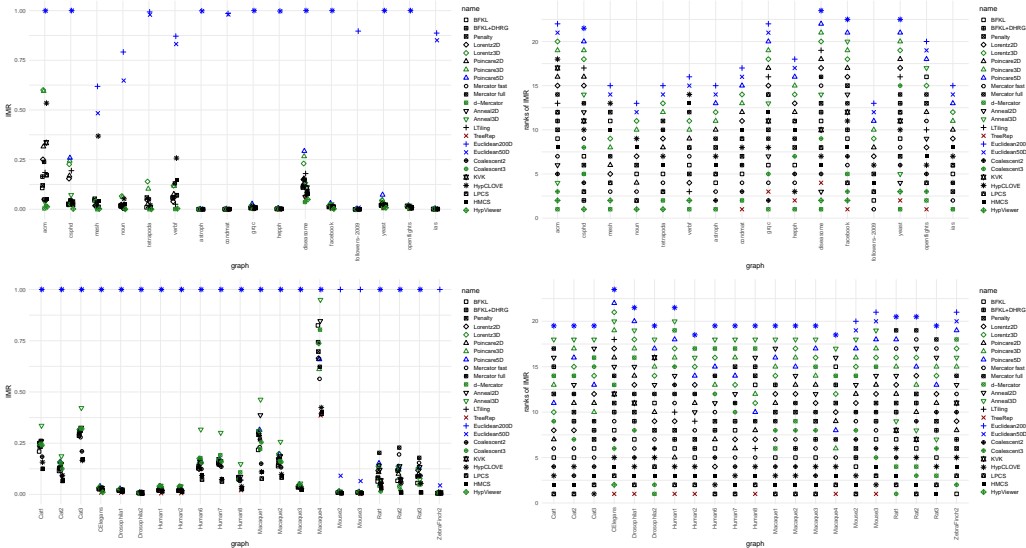

Figure 8: Quality assessment of embedders on real-world hierarchies and networks: -log(MR).

The discrepancy in the result of Euclidean higher-dimensional embeddings has been previously observed and studied in Bansal and Benton (2021); the reported values did arise as a result of using a different setting where the Euclidean embeddings were regularized[1]. Other differences in experimental results might be caused by differences in hyperparameters; the repository only gives the values of hyperparameters used to reproduce the 10-dimensional embeddings.

---

[1]https://github.com/facebookresearch/poincare-embeddings/issues/35

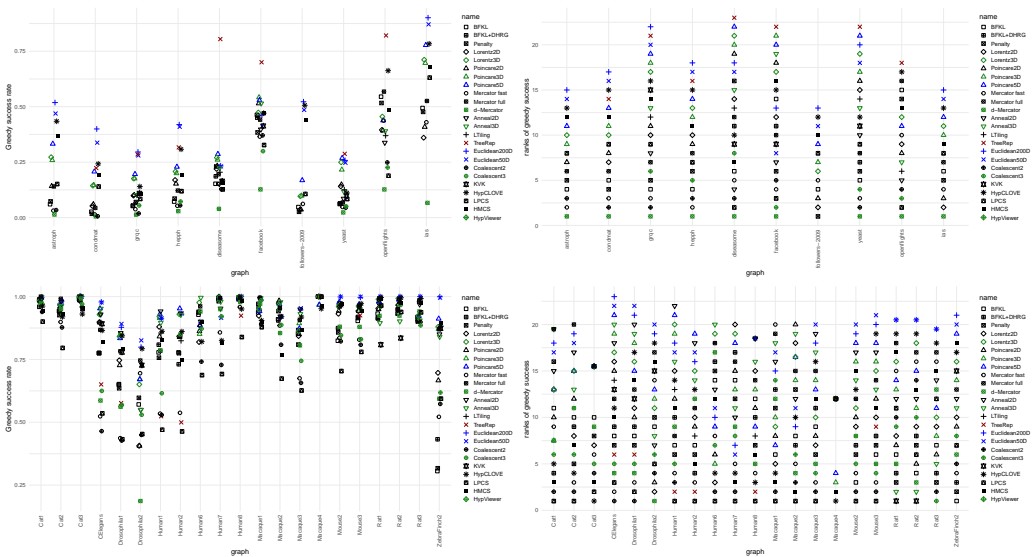

Figure 9: Quality assessment of embedders on real-world networks: greedy success rate (GSR).

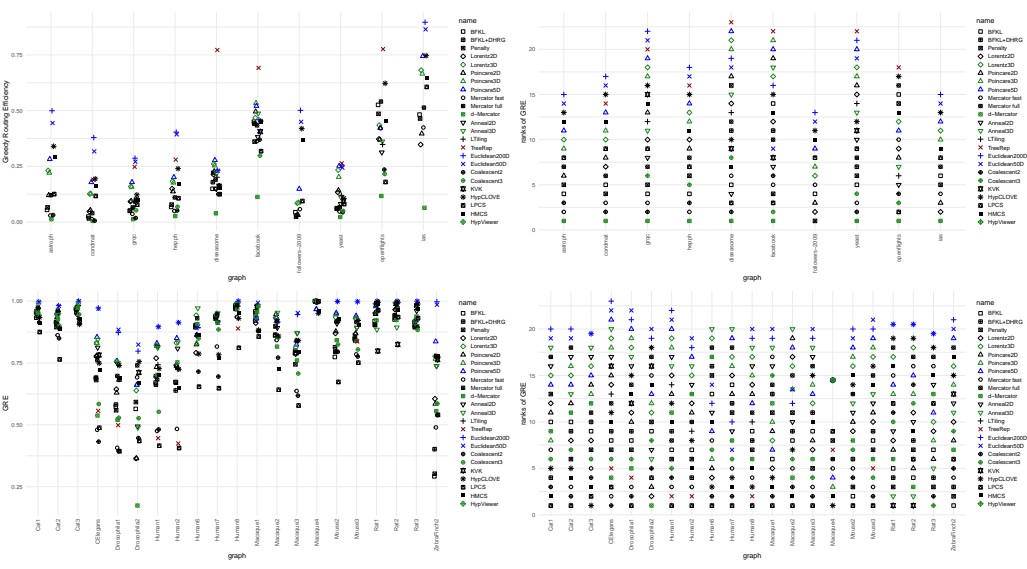

Figure 10: Quality assessment of embedders on real-world networks: greedy routing efficiency (GRE).

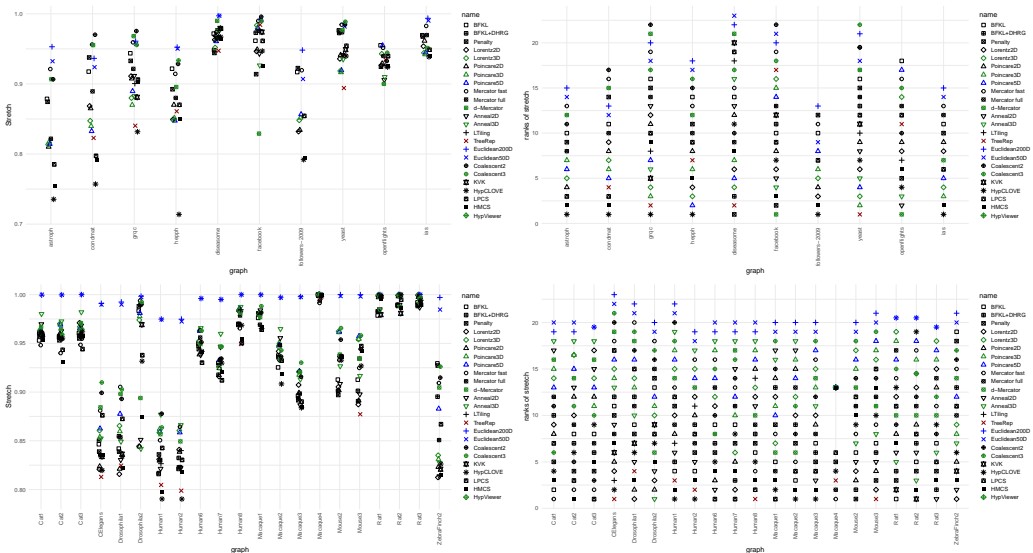

Figure 11: Quality assessment of embedders on real-world networks: -log(GSF).

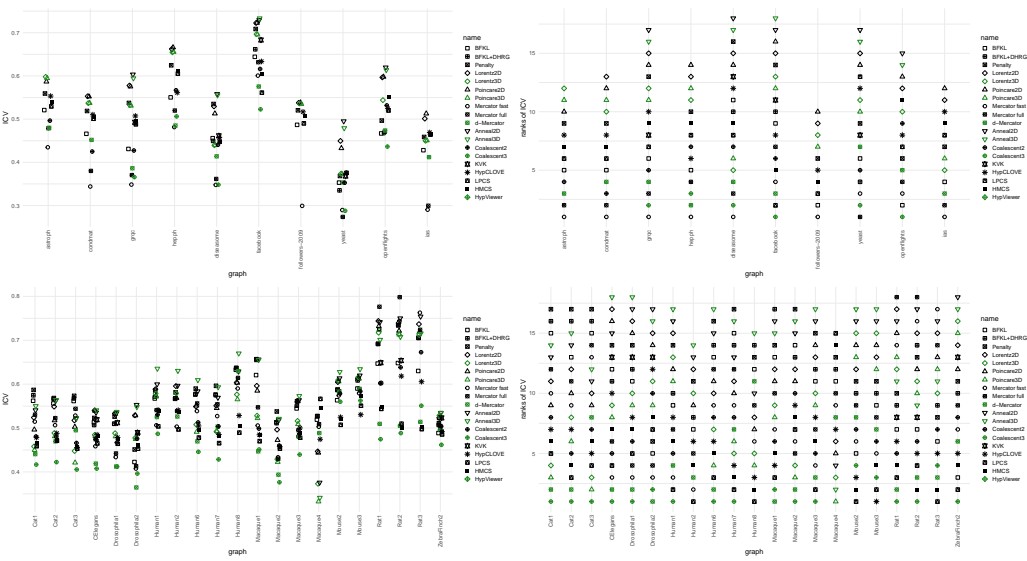

Figure 12: Quality assessment of embedders on real-world networks: information control value (ICV).

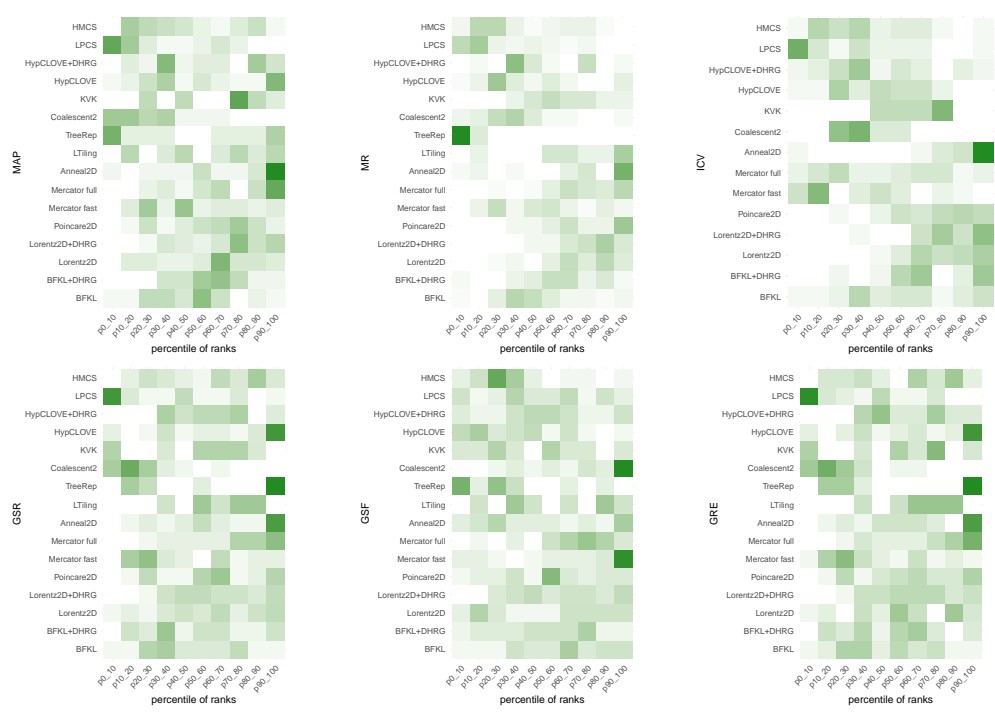

Figure 13: Aggregates for MR, ICV, GSR, and GSF measures on real-world networks and hierarchies.

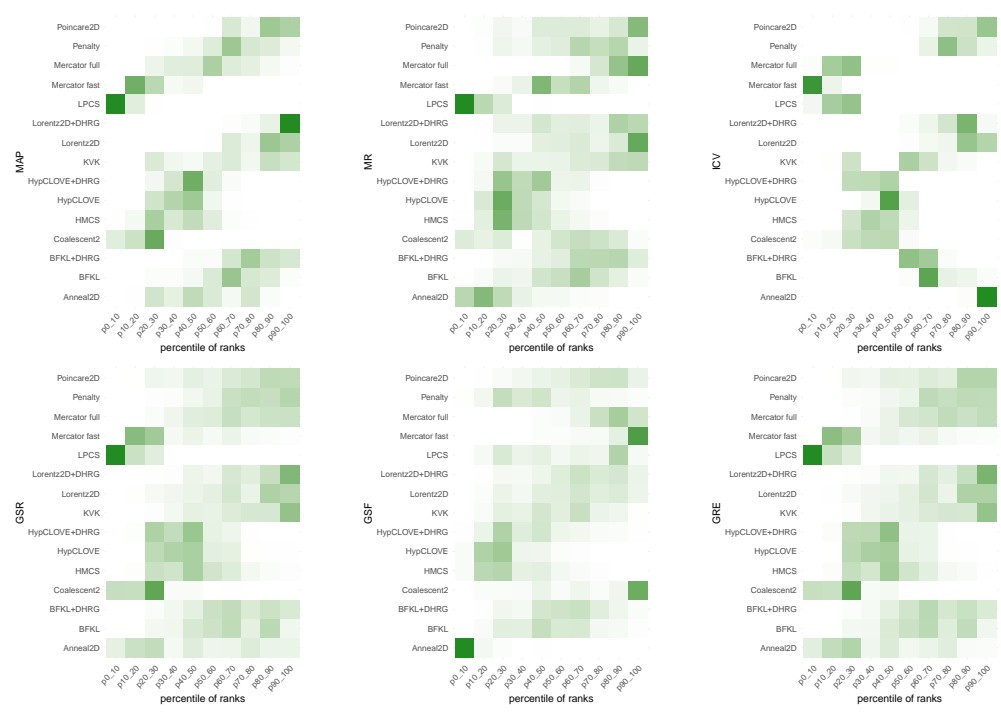

Figure 14: Aggregates for MR, ICV, GSR, and GSF measures on simulated networks.

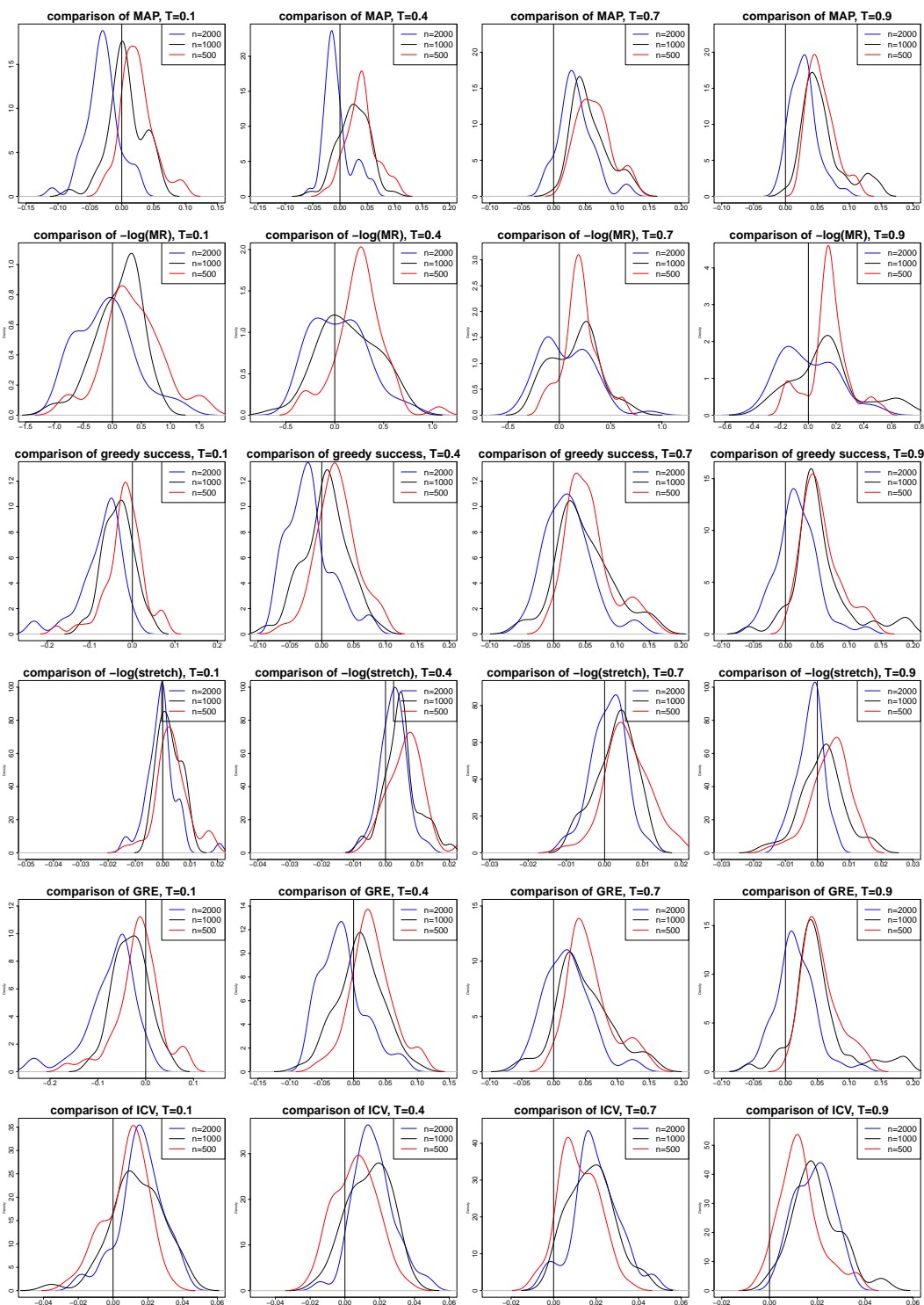

Figure 15: Density plots of the differences between the values of quality measures obtained by Lorentz 2D and BFKL embedders. Top to bottom: MAP, -log(MR), GSR, -log(GSF), GRE, ICV. Left to right: $T = 0.1$, $T = 0.4$, $T = 0.7$, $T = 0.9$. Negative values indicate that BFKL embedder performed better.

| | MAP | | MR | | GSR | | GSF | | ICV | |
|---|---|---|---|---|---|---|---|---|---|---|
| | Coeff. | Pr($>\lvert z\rvert$) | Coeff. | Pr($>\lvert z\rvert$) | Coeff. | Pr($>\lvert z\rvert$) | Coeff. | Pr($>\lvert z\rvert$) | Coeff. | Pr($>\lvert z\rvert$) |
| Intercept | -1.9701 | 7.09e-08 | -1.4818 | 2.32e-09 | 0.68510 | 0.007420 | -1.2612 | 1.28e-07 | 43.3744 | 9.77e-14 |
| Temp=0.4 | -0.8881 | 0.003800 | -0.3620 | 0.159600 | -2.1156 | 3.12e-12 | -1.1232 | 5.73e-05 | -1.4432 | 0.000874 |
| Temp=0.7 | -4.4173 | 5.20e-14 | -0.5747 | 0.028400 | -3.9175 | <2e-16 | -0.6677 | 0.010650 | -4.1914 | 8.69e-10 |
| Temp=0.9 | -4.4173 | 5.20e-14 | -0.2939 | 0.251100 | -4.3845 | <2e-16 | 0.15180 | 0.538150 | -4.6607 | 1.99e-07 |
| Size = 1000 | 1.6316 | 7.70e-05 | 1.0513 | 3.07e-05 | 0.92710 | 0.001930 | 0.76950 | 0.001960 | 2.37400 | 7.85e-05 |
| Size = 2000 | 3.7975 | <2e-16 | 1.8908 | 1.95e-14 | 2.53030 | 2.84e-15 | 1.58290 | 6.58e-11 | 6.30360 | 1.13e-08 |
| $R_{BFKL}/R_{L2}$ | — | — | — | — | — | — | — | — | -53.9572 | 5.63e-14 |
| N | 600 | | 600 | | 600 | | 600 | | 600 | |
| ACC$_{cv}$ | 0.8835 | | 0.6917 | | 0.8231 | | 0.7266 | | 0.9266 | |
| ACC$_{bench}$ | 0.7867 | | 0.6683 | | 0.6200 | | 0.6833 | | 0.8967 | |
| $\kappa$ | 0.6165 | | 0.2486 | | 0.6133 | | 0.2717 | | 0.5011 | |

Table 6: Results of logit regressions for the determinants of BFKL embedder outperforming Lorentz 2D embedder in terms of quality measures. ACC$_{cv}$ and $\kappa$ are average accuracy and Kappa from 20-fold cross-validation; ACC$_{bench}$ is the accuracy of the naive model (always predict mode). Results for GRE in Table 1.

| graph name | NOUN | VERB | ASTRO | COND | GRQC | HEPPH |
|---|---|---|---|---|---|---|
| Poincaré 2D MR (ours) | 88.0 | 15.7 | 1127.0 | 889.4 | 68.8 | 302.2 |
| Poincaré 2D MR | 90.7 | 10.7 | — | — | — | — |
| Poincaré 2D MAP (ours) | 0.105 | 0.314 | 0.324 | 0.391 | 0.660 | 0.472 |
| Poincaré 2D MAP | 0.118 | 0.365 | — | — | — | — |
| Lorentz 2D MR (ours) | 43.0 | 42.1 | 1104.8 | 949.2 | 81.4 | 293.2 |
| Lorentz 2D MR | 22.8 | 3.64 | — | — | — | — |
| Lorentz 2D MAP (ours) | 0.168 | 0.184 | 0.306 | 0.345 | 0.599 | 0.417 |
| Lorentz 2D MAP | 0.305 | 0.579 | — | — | — | — |
| Euclidean 50D MR (ours) | 1.5 | 1.2 | 1.0 | 1.0 | 1.0 | 1.0 |
| Euclidean 50D MR | 1281.7 | — | — | — | — | — |
| Euclidean 50D MAP (ours) | 0.921 | 0.908 | 0.988 | 0.968 | 1.000 | 0.980 |
| Euclidean 50D MAP | 0.140 | — | 0.376 | 0.356 | 0.522 | 0.434 |

Table 7: Our results compared with the results from [Nickel and Kiela, 2017; Nickel and Kiela, 2018].

## H    REPEATED EXPERIMENTS

In Tables 8 and 9 we list the results of repeated experiments on the NOUN hierarchy and the GRQC, YEAST, MOUSE3, HUMAN1, DROSOPHILA1 and CELEGANS networks. In most cases, the differences are minor and do not affect the rankings.

## I    TRIVIA ABOUT THE NOUN DATASET

This Appendix gives details about our experiments with the NOUN dataset, i.e., the WordNet hypernymy structure. This was the first hierarchy that PE/LE have been benchmarked on, common in ML studies.

We get MAP of 0.284 using BFKL which is significantly better than the result of Poincaré 2D of 0.118, but not the result of Lorentz 2D of 0.305, according to Nickel and Kiela (2018). However, our results are different: 0.105 for 2D PE and 0.168 for 2D LE. Furthermore, while the PE/LE papers mention the good performance of their embedding methods, on our machine, BFKL is almost 100 times faster than LE, which is especially impressive given that BFKL runs on a single CPU. Furthermore, the DHRG improvement improves the BFKL embedding from dMAP 0.050 to dMAP 0.411, while LE is improved from dMAP 0.192 to dMAP 0.320. This suggests that the layered approach of BFKL produces a better structure of the embedding. Furthermore, the combination of BFKL+DHRG is still more than 10 times faster than 2D LE. (The dMAP result of 0.050 is very low compared to the continuous result of 0.284; this seems to be an outlier, in our other experiments the results of MAP and dMAP are very similar.)

| | graph name | grqc | yeast | CElegs | Human1 | Droso1 | Mouse3 |
|---|---|---|---|---|---|---|---|
| | nodes | 4158 | 1458 | 279 | 493 | 350 | 1076 |
| | edges (undirected) | 13422 | 1948 | 2287 | 7773 | 2887 | 90811 |
| GSR | BFKL | 0.059 (0.055,0.065) | 0.062 (0.060,0.064) | 0.781 (0.770,0.792) | 0.779 (0.742,0.804) | 0.629 (0.618,0.642) | 0.905 (0.903,0.908) |
| | BFKL + DHRG | - | - | 0.777 (0.765,0.789) | 0.810 (0.780,0.855) | 0.638 (0.629,0.646) | 0.921 (0.918,0.925) |
| | Poincare 2D | - | 0.146 (0.132,0.154) | 0.894 (0.879,0.900) | 0.864 (0.845,0.880) | 0.733 (0.721,0.748) | 0.942 (0.936,0.946) |
| | Poincare 3D | - | 0.232 (0.221,0.245) | 0.939 (0.930,0.950) | 0.917 (0.913,0.922) | 0.830 (0.819,0.841) | 0.968 (0.965,0.970) |
| | Lorentz 2D | 0.107 (0.102,0.114) | 0.137 (0.131,0.143) | 0.889 (0.876,0.895) | 0.882 (0.855,0.907) | 0.743 (0.733,0.752) | 0.938 (0.935,0.941) |
| | Lorentz 2D + DHRG | 0.128 (0.120,0.135) | 0.135 (0.130,0.141) | 0.840 (0.834,0.845) | 0.875 (0.852,0.904) | 0.678 (0.666,0.690) | 0.925 (0.922,0.930) |
| | Lorentz 3D | - | 0.240 (0.234,0.245) | 0.921 (0.914,0.928) | 0.922 (0.911,0.930) | 0.834 (0.829,0.840) | 0.966 (0.964,0.969) |
| | penalty | 0.058 (0.054,0.065) | 0.062 (0.058,0.064) | 0.782 (0.771,0.799) | 0.785 (0.762,0.807) | 0.622 (0.603,0.637) | 0.915 (0.908,0.919) |
| | anneal2 | 0.076 (0.073,0.077) | 0.087 (0.083,0.091) | 0.924 (0.913,0.928) | 0.918 (0.898,0.932) | 0.841 (0.830,0.851) | 0.968 (0.966,0.970) |
| | anneal3 | 0.091 (0.085,0.098) | 0.119 (0.113,0.125) | 0.931 (0.923,0.945) | 0.948 (0.908,0.964) | 0.849 (0.836,0.864) | 0.950 (0.947,0.952) |
| | KVK | 0.119 (0.105,0.131) | 0.099 (0.098,0.100) | 0.871 (0.860,0.884) | - | 0.794 (0.784,0.814) | - |
| | CLOVE | 0.158 (0.146,0.166) | 0.105 (0.094,0.109) | 0.844 (0.831,0.857) | 0.831 (0.813,0.850) | 0.829 (0.819,0.838) | 0.968 (0.966,0.970) |
| | LPCS | 0.085 (0.081,0.087) | 0.090 (0.086,0.095) | 0.518 (0.494,0.553) | 0.526 (0.489,0.559) | 0.432 (0.417,0.443) | 0.760 (0.745,0.774) |
| | HMCS | 0.106 (0.105,0.108) | - | 0.818 (0.806,0.825) | 0.804 (0.769,0.830) | 0.784 (0.754,0.796) | 0.939 (0.934,0.944) |
| | Mercator fast | - | 0.048 (0.048,0.049) | 0.524 (0.523,0.525) | 0.534 (0.532,0.535) | 0.440 (0.436,0.445) | 0.830 (0.829,0.832) |
| | Mercator full | - | 0.068 (0.067,0.070) | 0.845 (0.830,0.860) | 0.797 (0.788,0.811) | 0.758 (0.744,0.772) | 0.960 (0.959,0.961) |
| | d-Mercator | 0.015 (0.014,0.016) | 0.021 (0.019,0.023) | 0.582 (0.579,0.585) | 0.788 (0.787,0.789) | 0.563 (0.562,0.566) | 0.881 (0.880,0.881) |
| GRE | BFKL | 0.056 (0.053,0.062) | 0.061 (0.059,0.063) | 0.692 (0.685,0.699) | 0.674 (0.643,0.695) | 0.561 (0.553,0.569) | 0.850 (0.849,0.852) |
| | BFKL + DHRG | - | - | 0.689 (0.681,0.694) | 0.698 (0.672,0.735) | 0.571 (0.564,0.577) | 0.857 (0.853,0.862) |
| | Poincare 2D | - | 0.139 (0.126,0.147) | 0.776 (0.763,0.782) | 0.750 (0.738,0.761) | 0.633 (0.624,0.644) | 0.869 (0.864,0.872) |
| | Poincare 3D | - | 0.217 (0.208,0.231) | 0.835 (0.829,0.842) | 0.817 (0.812,0.819) | 0.745 (0.736,0.754) | 0.937 (0.934,0.938) |
| | Lorentz 2D | 0.099 (0.095,0.105) | 0.131 (0.126,0.136) | 0.771 (0.762,0.776) | 0.768 (0.747,0.786) | 0.644 (0.637,0.650) | 0.865 (0.861,0.866) |
| | Lorentz 2D + DHRG | 0.117 (0.110,0.122) | 0.129 (0.124,0.134) | 0.737 (0.733,0.744) | 0.761 (0.743,0.784) | 0.597 (0.587,0.607) | 0.854 (0.851,0.858) |
| | Lorentz 3D | - | 0.225 (0.219,0.230) | 0.824 (0.818,0.829) | 0.820 (0.813,0.825) | 0.750 (0.745,0.756) | 0.935 (0.933,0.937) |
| | penalty | 0.055 (0.052,0.062) | 0.061 (0.057,0.062) | 0.686 (0.677,0.701) | 0.683 (0.661,0.699) | 0.546 (0.529,0.558) | 0.842 (0.837,0.845) |
| | anneal2 | 0.070 (0.068,0.071) | 0.083 (0.080,0.087) | 0.807 (0.799,0.811) | 0.798 (0.784,0.809) | 0.735 (0.728,0.741) | 0.894 (0.893,0.896) |
| | anneal3 | 0.084 (0.078,0.089) | 0.113 (0.108,0.118) | 0.825 (0.816,0.837) | 0.860 (0.825,0.876) | 0.752 (0.741,0.767) | 0.891 (0.889,0.893) |
| | KVK | 0.108 (0.095,0.118) | 0.095 (0.094,0.096) | 0.765 (0.756,0.775) | - | 0.701 (0.692,0.716) | - |
| | CLOVE | 0.136 (0.127,0.142) | 0.100 (0.090,0.104) | 0.738 (0.727,0.745) | 0.701 (0.692,0.719) | 0.727 (0.718,0.735) | 0.918 (0.916,0.920) |
| | LPCS | 0.079 (0.076,0.081) | 0.087 (0.082,0.091) | 0.474 (0.454,0.504) | 0.458 (0.430,0.485) | 0.393 (0.381,0.404) | 0.734 (0.721,0.747) |
| | HMCS | 0.098 (0.097,0.100) | - | 0.720 (0.710,0.733) | 0.682 (0.653,0.702) | 0.681 (0.654,0.691) | 0.901 (0.895,0.907) |
| | Mercator fast | - | 0.048 (0.047,0.048) | 0.481 (0.480,0.482) | 0.475 (0.474,0.476) | 0.410 (0.406,0.414) | 0.786 (0.785,0.787) |
| | Mercator full | - | 0.066 (0.065,0.068) | 0.744 (0.731,0.756) | 0.693 (0.685,0.705) | 0.679 (0.668,0.690) | 0.916 (0.915,0.916) |
| | d-Mercator | 0.014 (0.014,0.016) | 0.021 (0.019,0.023) | 0.533 (0.529,0.536) | 0.704 (0.703,0.705) | 0.522 (0.521,0.524) | 0.840 (0.839,0.840) |

Table 8: Repeated experiments: greedy routing measures. Mean values from 5 runs. Bootstrapped confidence intervals in brackets.

| | graph name | grqc | yeast | CElegs | Human1 | Droso1 | Mouse3 |
|---|---|---|---|---|---|---|---|
| | nodes | 4158 | 1458 | 279 | 493 | 350 | 1076 |
| | edges (undirected) | 13422 | 1948 | 2287 | 7773 | 2887 | 90811 |
| MAP | BFKL | 0.492 (0.484,0.505) | 0.735 (0.725,0.746) | 0.462 (0.456,0.469) | 0.568 (0.550,0.579) | 0.384 (0.379,0.388) | 0.562 (0.559,0.564) |
| | BFKL + DHRG | - | - | 0.464 (0.460,0.467) | 0.575 (0.560,0.587) | 0.391 (0.384,0.397) | 0.577 (0.574,0.578) |
| | Poincare 2D | - | 0.671 (0.641,0.682) | 0.493 (0.482,0.498) | 0.640 (0.634,0.645) | 0.387 (0.384,0.390) | 0.575 (0.572,0.577) |
| | Poincare 3D | - | 0.765 (0.749,0.771) | 0.576 (0.573,0.580) | 0.719 (0.716,0.722) | 0.482 (0.476,0.486) | 0.652 (0.651,0.653) |
| | Lorentz 2D | 0.612 (0.605,0.615) | 0.535 (0.525,0.540) | 0.491 (0.486,0.496) | 0.645 (0.637,0.651) | 0.390 (0.387,0.394) | 0.573 (0.571,0.574) |
| | Lorentz 2D + DHRG | 0.745 (0.741,0.747) | 0.782 (0.764,0.795) | 0.481 (0.477,0.487) | 0.637 (0.628,0.646) | 0.383 (0.374,0.391) | 0.584 (0.583,0.586) |
| | Lorentz 3D | - | 0.764 (0.758,0.770) | 0.574 (0.572,0.575) | 0.715 (0.711,0.719) | 0.489 (0.481,0.493) | 0.653 (0.652,0.654) |
| | penalty | 0.470 (0.464,0.478) | 0.733 (0.721,0.746) | 0.451 (0.444,0.457) | 0.577 (0.561,0.587) | 0.370 (0.361,0.376) | 0.574 (0.571,0.575) |
| | anneal2 | 0.613 (0.611,0.614) | 0.644 (0.638,0.649) | 0.537 (0.536,0.539) | 0.670 (0.668,0.673) | 0.479 (0.473,0.484) | 0.610 (0.607,0.611) |
| | anneal3 | 0.648 (0.644,0.651) | 0.675 (0.671,0.678) | 0.585 (0.575,0.589) | 0.800 (0.789,0.810) | 0.515 (0.512,0.520) | 0.652 (0.646,0.653) |
| | KVK | 0.639 (0.574,0.663) | 0.750 (0.742,0.756) | 0.489 (0.483,0.494) | - | 0.432 (0.423,0.438) | - |
| | CLOVE | 0.684 (0.683,0.685) | 0.775 (0.772,0.777) | 0.460 (0.458,0.465) | 0.561 (0.546,0.571) | 0.415 (0.409,0.422) | 0.466 (0.461,0.473) |
| | LPCS | 0.558 (0.557,0.561) | 0.652 (0.644,0.661) | 0.271 (0.255,0.296) | 0.360 (0.344,0.376) | 0.212 (0.200,0.224) | 0.397 (0.387,0.407) |
| | HMCS | 0.616 (0.614,0.619) | - | 0.451 (0.444,0.459) | 0.566 (0.557,0.572) | 0.385 (0.378,0.390) | 0.533 (0.531,0.537) |
| | Mercator fast | - | 0.682 (0.680,0.684) | 0.337 (0.336,0.337) | 0.411 (0.411,0.412) | 0.270 (0.270,0.270) | 0.520 (0.520,0.520) |
| | Mercator full | - | 0.753 (0.752,0.755) | 0.486 (0.482,0.494) | 0.549 (0.548,0.551) | 0.423 (0.418,0.431) | 0.585 (0.584,0.585) |
| | d-Mercator | 0.317 (0.314,0.319) | 0.215 (0.206,0.223) | 0.370 (0.367,0.372) | 0.636 (0.635,0.637) | 0.346 (0.342,0.350) | 0.583 (0.583,0.583) |
| | orig TreeRep rec | - | - | 0.199 (0.191,0.207) | 0.264 (0.249,0.277) | 0.233 (0.223,0.242) | 0.236 (0.227,0.241) |
| | orig TreeRep norec | - | - | 0.220 (0.207,0.233) | 0.263 (0.254,0.276) | 0.233 (0.228,0.244) | 0.243 (0.232,0.254) |
| MR | BFKL | 171.1 (161.1,181.5) | 53.4 (51.5,56.5) | 37.5 (36.7,38.7) | 50.1 (48.6,50.9) | 53.0 (52.4,53.8) | 102.1 (101.5,103.3) |
| | BFKL + DHRG | - | - | 36.9 (36.0,37.6) | 48.5 (46.5,49.9) | 49.9 (49.6,50.4) | 98.9 (98.4,99.4) |
| | Poincare 2D | - | 32.7 (30.7,33.5) | 31.4 (31.1,31.7) | 42.8 (39.9,49.1) | 47.1 (46.6,47.9) | 96.5 (96.2,96.9) |
| | Poincare 3D | - | 22.1 (21.5,22.7) | 27.3 (26.9,27.8) | 25.7 (25.3,26.4) | 39.6 (39.2,39.8) | 84.3 (84.1,84.5) |
| | Lorentz 2D | 75.3 (72.4,79.1) | 38.3 (37.9,38.9) | 31.6 (31.4,31.8) | 40.5 (39.2,43.4) | 47.2 (47.0,47.4) | 96.6 (96.3,96.9) |
| | Lorentz 2D + DHRG | 84.5 (80.3,87.5) | 33.4 (32.8,34.2) | 32.6 (32.5,32.7) | 40.5 (39.4,42.6) | 47.9 (47.7,48.0) | 96.4 (96.1,96.8) |
| | Lorentz 3D | - | 21.6 (20.9,22.4) | 27.2 (27.0,27.3) | 26.5 (25.6,27.9) | 39.3 (38.8,39.9) | 84.0 (83.9,84.1) |
| | penalty | 143.0 (130.6,152.0) | 56.3 (53.0,64.2) | 36.2 (35.3,37.2) | 45.6 (44.0,46.3) | 49.8 (49.3,50.3) | 97.1 (96.7,97.6) |
| | anneal2 | 156.8 (152.7,160.3) | 74.2 (71.2,76.7) | 33.2 (32.5,33.5) | 42.2 (40.9,43.3) | 46.4 (46.3,46.5) | 93.8 (93.6,94.0) |
| | anneal3 | 121.0 (116.4,127.0) | 59.2 (55.9,62.1) | 27.3 (26.9,28.0) | 19.3 (18.1,22.2) | 38.4 (38.0,38.9) | 80.2 (79.6,81.8) |
| | KVK | 130.1 (125.3,141.2) | 49.6 (48.1,50.7) | 36.1 (35.2,36.9) | - | 47.4 (47.2,47.6) | - |
| | CLOVE | 158.2 (153.9,165.4) | 49.4 (48.6,50.9) | 42.9 (42.0,43.6) | 55.6 (52.6,58.7) | 63.0 (61.7,64.1) | 164.9 (161.0,167.9) |
| | LPCS | 119.8 (116.9,122.6) | 47.8 (46.9,48.7) | 50.7 (48.9,52.0) | 63.4 (61.6,65.0) | 73.2 (72.6,73.7) | 157.7 (152.5,161.9) |
| | HMCS | 119.6 (118.2,120.9) | - | 43.6 (42.6,45.1) | 54.6 (53.5,55.4) | 68.2 (67.3,69.4) | 133.1 (128.5,134.9) |
| | Mercator fast | - | 50.8 (50.7,50.9) | 37.8 (37.7,37.8) | 41.5 (41.5,41.6) | 54.3 (54.3,54.4) | 103.5 (103.5,103.5) |
| | Mercator full | - | 45.7 (45.4,46.0) | 34.2 (34.1,34.3) | 41.2 (41.1,41.3) | 47.8 (47.6,47.9) | 99.5 (99.4,99.6) |
| | d-Mercator | 500.6 (478.4,509.2) | 187.8 (174.2,201.7) | 34.7 (34.3,35.0) | 24.1 (24.1,24.1) | 41.0 (40.9,41.1) | 92.8 (92.8,92.8) |

Table 9: Repeated experiments: MAP and MR measures. Mean values from 5 runs. Bootstrapped confidence intervals in brackets.

| embedder | MAP | dMAP | MR | dMR |
|---|---|---|---|---|
| BFKL | 0.280 (0.276,0.285) | 0.049 (0.048,0.049) | 64.7 (63.3,66.0) | 807.5 (785.4,825.9) |
| BFKL + DHRG | | 0.442 (0.423,0.456) | | 34.9 (33.9,37.3) |
| Poincare 2D | 0.105 (0.104,0.105) | 0.018 (0.018,0.018) | 89.5 (88.1,90.7) | 2536.2 (2498.8,2565.1) |
| Poincare 2D + DHRG | | 0.062 (0.062,0.063) | | 370.3 (351.7,388.9) |
| Poincare 3D | 0.490 (0.487,0.492) | | 16.1 (15.7,16.3) | |
| Lorentz 2D | 0.196 (0.195,0.196) | 0.193 (0.191,0.193) | 41.8 (41.2,42.3) | 42.0 (41.4,42.4) |
| Lorentz 2D + DHRG | 0.326 (0.323,0.328) | 0.324 (0.321,0.326) | 30.1 (29.2,30.8) | 28.4 (27.6,29.0) |
| Lorentz 3D | 0.506 (0.506,0.507) | | 14.8 (14.7,15.0) | |
| CLOVE | 0.769 (0.763,0.777) | 0.630 (0.624,0.636) | 18.8 (18.3,19.7) | 23.1 (22.6,24.1) |
| CLOVE + DHRG | | 0.783 (0.779,0.787) | | 20.9 (20.5,21.6) |

Table 10: Repeated experiments on the NOUN hierarchy. Mean values from 5 runs. Bootstrapped confidence intervals in brackets.

This is a very large hierarchy, so it is not feasible to run slower embedders on it. We have also run the new CLOVE embedder, which achieves MAP of 0.769, which is significantly better than the earlier two-dimensional embedders. Furthermore, it runs over 3 times faster than BFKL. It is possible to apply the DHRG imrpovement to this embedding, obtaining an even better value of MAP (0.791).

These results are consistent across multiple runs (Table 10).

