# OpenReview forum: "Bridging ML and algorithms: comparison of hyperbolic embeddings"
_ICLR.cc/2026/Conference — ICLR 2026 Poster_

### Official Review · Reviewer_XXeR · 2025-10-28

**Soundness:** 3
**Presentation:** 3
**Contribution:** 2
**Rating:** 4
**Confidence:** 3

**Summary:**

The paper systematically compares hyperbolic embedding methods spanning three communities. The headline result is that the BFKL algorithm is typically ~100× faster than Poincaré (NIPS’17) and Lorentz (ICML’18) embeddings while achieving similar quality under MAP/MR and greedy routing metrics. The authors also introduce an Information Control Value (ICV) criterion to penalize excessive radius/dimension, arguing that apparent gains from higher dimension are often optimization artifacts.

**Strengths:**

1. The work fills a real gap by bridging communities that have studied hyperbolic embeddings in isolation, delivering a broad, apples-to-apples experimental comparison across many methods, datasets, and metrics.
2. The paper thoughtfully treats numerical stability and contributes the ICV criterion to counter dimension/radius inflation, leading to more balanced method selection. The inclusion of both real hierarchies and scale-free networks, plus synthetic HRG controls with regression analyses on temperature/size, strengthens the empirical narrative.

**Weaknesses:**

1. There is no theoretical support in the paper.
2. Figures 2 and 5 are difficult to read.
3. Despite the breadth, some baseline coverage choices limit conclusions, which may bias the landscape.
4. Some evaluations rely on discrete variants (dMAP/dMR) to avoid precision issues, which can shift scores relative to standard definitions.

**Questions:**

No questions

---

> ### Author Response · Authors · 2025-11-26
>
> > There is no theoretical support in the paper.
>
> Theoretical support is not the purpose of the paper. We are planning to write an extended paper on the theoretical support for the ICV measure (there is not enough space for this in the paper under review). At the same time, the papers we cite include analyses of the theoretical properties of their models or measures. Our comparison is based on material with a theoretical background.
>
> > Some evaluations rely on discrete variants (dMAP/dMR) to avoid precision issues, which can shift scores relative to standard definitions.
>
> We mostly use the discrete variants to evaluate the DHRG method, which relies on them; this method improves other embeddings, so we apply it to the Poincaré and BFKL embeddings. We also note that these discrete variants can be computed faster than their continuous counterparts. Nevertheless, despite the fast computation, we do not use these measures for other algorithms.
>
> > Despite the breadth, some baseline coverage choices limit conclusions, which may bias the landscape.
>
> We kindly ask for a clarification on this issue. Which baseline coverage choices limit conclusions?

---

### Official Review · Reviewer_VCXG · 2025-10-30

**Soundness:** 2
**Presentation:** 2
**Contribution:** 1
**Rating:** 2
**Confidence:** 4

**Summary:**

The paper presents more like a survey with experiments bridging hyperbolic embedding methods in machine learning and classical network embedding algorithms. It reviews learning-based hyperbolic embeddings algorithms commonly used in ML with embeddings algorithms from network theory. The authors argue that the ML community often focuses on gradient-based methods, overlooking algorithmic approaches from network theory, that could show better efficiency with comparable performances.

**Strengths:**

The paper highlights the diversity of hyperbolic embedding methods and draws attention to classical embedding algorithms from network theory. Conducted experiments for both sides algorithms and draw comparison.

It makes the point that representations from network theory is more efficient and could even performs better than ML learning-based algorithms tracing from Nickel and Kiela (2017, 2018) and thus should be favored, cited and adopted.

The survey-style overview may help readers unfamiliar with the historical algorithmic side of hyperbolic embeddings algorithms.

**Weaknesses:**

The criticism that ML work “fails to cite/use” algorithmic hyperbolic embedding literature is not very compelling.

- The reason the ML community focuses more on works tracing from Nickel and Kiela (2017, 2018) is that, they allow gradient flow to the embedding layer, and trainable, end-to-end representations that adapt to downstream tasks, not just for embedding data (rather the reported performances serves as a demo of the learnt representation). Static embeddings from algorithmic approaches (network theory) cannot serve this role. Ultimately in my opinion, the hyperbolic learning community targets at an end-to-end hyperbolic networks that can replace existing Euclidean-based networks on some tasks. And it's already well-established how important learnable embedding layers are.

- The focus on metrics is also somewhat misplaced, since for downstream tasks (e.g., node or graph classification), the rank and precision of distances is very relevant to downstream performance, that truly matters, especially considering current e.g. classification algorithms are largely dependent on the hyperbolic distance.

- Some reported results appear unreliable—for example, the 2D Poincaré and Lorentz embeddings from Nickel and Kiela (2017, 2018) were not reproduced, even though they were already shown in many following ML works, can be reproduced.

- The framing of the paper is also misleading: it reads more like a survey rather than a genuine attempt to bridge ML and algorithms due to the reasons stated above.

**Questions:**

- How do the authors envision static embeddings from network theory adapting to downstream ML tasks that require trainable representations?

- what causes the non-reproducibility of Poincaré/Lorentz experiment results?

---

> ### Author Response · Authors · 2025-11-26
>
> > Some reported results appear unreliable -- for example, the 2D Poincaré and Lorentz embeddings from Nickel and Kiela (2017, 2018) were not reproduced, even though they were already shown in many following ML works, can be reproduced.
>
> We would like to clarify that by "reproducing," we mean not only citing the work (as in Sala et al.), but independently obtaining the same experimental results. Unfortunately, we were unable to achieve this due to insufficient information provided in the original papers. We would greatly appreciate further clarification regarding which specific machine learning works the Reviewer refers to when mentioning reproduction (as opposed to citation). Such information would be invaluable to us, as we believe there may be key details we are currently missing.
>
> Of course, these embedders work, which has been reproduced (including by us). However, we were unable to reproduce the poor performance of high-dimensional Euclidean embeddings.  During the revision phase we have discovered that other researchers failed to reproduce this as well (see the paper https://aclanthology.org/2021.insights-1.8/ which we cite in the current revision). Also, we failed to reproduce the significantly better performance of Lorentz embeddings over Poincaré embeddings (we have noticed that the paper "Tree! I am no Tree! I am a Low Dimensional Hyperbolic Embedding" by Sonthalia and Gilbert does include a comparison of these methods, and LE does not appear better than PE according to their results). Other issues, such as the better results of Lorentz 2D on the noun hierarchy, could be explained by using different hyperparameter values, so we consider it a relatively minor issue. In the repo, the authors mention that one can use the hyperparameters included in the source code to reproduce 10D results; since we focus on lower-dimensional embeddings, we have not experimented with 10D, and we copied these hyperparameters for other numbers
> of dimensions. At the same time, we stress that the hyperparameters for all their experimental results should be included for reproducibility.)
>
> > The criticism that ML work “fails to cite/use” algorithmic hyperbolic embedding literature is not very compelling.
>
> We have noticed that a significant share of current papers directly claim that NK17 were the first to consider hyperbolic embeddings or that they were the first to show that hierarchies are well-modelled by hyperbolic geometry (we mention one in the current revision, we have also seen a lot of such papers in the NEGEL workshop at NeurIPS and the Beyond Euclidean Workshop on ICCV, although we have looked at these workshops after the ICLR submission deadline, and it is hard to "cite" these workshops to justify our claim. Those claims are simply untrue. We believe this is a serious issue -- it spreads misinformation. Even if we omit the classical works in visualization or network theory, ML usage in embedders was started by at least Cannistraci's group -- they deserve fair mentioning here from the community.
>
> > The focus on metrics is also somewhat misplaced, since for downstream tasks (e.g., node or graph classification), the rank and precision of distances is very relevant to downstream performance, that truly matters, especially considering current e.g. classification algorithms are largely dependent on the hyperbolic distance.
>
> We do not deny the relevance of mAP and MeanRank measures for machine learning applications. We include both these measures, and measures popular in the NT community (such as the greedy routing statistics), in our experiments. According to our experiments, embedders which obtain good results on NT measures tend to also obtain good results on mAP and MeanRank (and vice versa).

---

> ### Author Response · Authors · 2025-11-26
>
> > The reason the ML community focuses more on works tracing from Nickel and Kiela (2017, 2018) is that, they allow gradient flow to the embedding layer, and trainable, end-to-end representations that adapt to downstream tasks, not just for embedding data (rather the reported performances serves as a demo of the learnt representation). Static embeddings from algorithmic approaches (network theory) cannot serve this role. Ultimately in my opinion, the hyperbolic learning community targets at an end-to-end hyperbolic networks that can replace existing Euclidean-based networks on some tasks. And it's already well-established how important learnable embedding layers are.
>
> > How do the authors envision static embeddings from network theory adapting to downstream ML tasks that require trainable representations?
>
> This seems to refer to the application of hyperbolic embeddings in GNNs. This is one specific application of hyperbolic embeddings (we generally do not talk much about applications in our paper, neither from the ML nor the NT side). Neither did Nickel and Kiela, who, as far as we see, did not explicitly foresee such an application. Up to our knowledge, for a given application, we can replace the embedder, or adjust it, keeping the original idea (as for example in the case of Mercator embedder, which has many variants). Nickel and Kiela's embedders also generate static embeddings. We are not aware of any issue that prevents replacing their embedder with some embedder coming from the network theory research (of course, after adapting the code). In particular, a recent paper https://arxiv.org/abs/2406.02772 uses the Mercator embedder in GNN research.

---

### Official Review · Reviewer_61kf · 2025-10-31

**Soundness:** 4
**Presentation:** 3
**Contribution:** 3
**Rating:** 8
**Confidence:** 5

**Summary:**

Hyperbolic embedding is a well-studied technique for graph representation learning. This paper presents a comprehensive comparison of algorithms from the ML/DL community and the traditional algorithm community. It presents a convincing survey that shows: the most popular methods derived by the ML/DL community (e.g. by Nickel and Kiela) are inferior to classic algorithms (e.g. by Blasius et al) in terms of both computational efficiency and embedding quality. This is a problem that many ML/DL researchers vaguely noticed but failed to explicitly point out. I highly appreciate the contributions of this work.

**Strengths:**

1. This paper presents a comprehensive and clear survey of both ML/DL approaches to hyperbolic embedding as well as traditional algorithmic approaches. The computational complexity of each approach is clearly discussed.

2. Very comprehensive experimental results to support the authors' claims.

3. Good writing styles -- the authors wrote a very clear preliminary section on hyperbolic geometry for the readers' information, followed by separate sections on algorithmic approaches and ML/DL approaches.

**Weaknesses:**

1. Some citations should be in brackets, e.g. on Line 42.

2. Figure 2, 3, and 5 are difficult to read -- the markers are too small and too similar. The same problem applies to the figures in the Appendix. If these figures are scaled to fit the page limit, I'd suggest re-design Table 1 to save some space instead of sacrificing the readability of your most illustrative figures.

3. Different hyperbolic embedding strategies perform differently on different structures, be it a tree, a deep but sparse graph, a shallow but dense graph, or a nearly fully-connected graph. You can refer to this ICLR workshop paper: https://arxiv.org/pdf/2407.16641?. The concept of local capacity explains why some methods work better on certain datasets. You may want to include more varied network datasets in your experiments, or generate datasets with distinct structures to test the methods.

**Questions:**

1. Line 288-290, the authors claim that "The hMDA method from (Sala et al., 2018) looks interesting, but it depends on the scaling factor, and it is not clear how to learn this parameter; therefore we do not include this method in our experiments". Do you mean the hMDS method? You need to explain more clearly why this method does apply to your experimental setting.

---

> ### Author Response · Authors · 2025-11-26
>
> Thanks for your comments! We have added the missing brackets.
>
> Indeed, there was a typo in the name of the method.
>
> The hMDS implementation (Julia file hmds-simple.jl) requires dataset, dimension, and scale parameters. We have managed to run this implementation on some of our graphs, and we did obtain embeddings, but we have observed that this was the case only when we guessed the value of the scale parameter reasonably well -- otherwise, no embedding was obtained (the program crashed), or the embedding was of a smaller dimension. As such, the behavior of this embedder was not stable enough for our comparison. So far, we were unable to find any method of setting a good value for this parameter safely in the paper or the repo; the 'hmds-runs.py' example simply uses a grid search, which would further increase the working time of the embedder. (We have not yet this explanation to the paper.)

---

### Official Review · Reviewer_1A68 · 2025-11-01

**Soundness:** 1
**Presentation:** 1
**Contribution:** 1
**Rating:** 2
**Confidence:** 5

**Summary:**

The paper aims to provide a systematic, cross-community comparison of hyperbolic embedding algorithms developed in machine learning (ML), network theory (NT), and algorithmic graph research.
The authors aim to close the gap between these communities by experimentally evaluating 14 embedding methods on 30 real-world and 450 simulated networks, comparing both embedding quality and computational efficiency.
This work aims to bridge ML and algorithmic perspectives on hyperbolic embeddings, attempting to show that older algorithmic methods are far more computationally efficient without sacrificing accuracy, and proposes ICV which is intended to be a fairer, theory-grounded quality metric for comparing embeddings.

**Strengths:**

The main strength of the paper is the mission to offer a wide comparison of algorithms for hyperbolic embedding of networks across domains.

**Weaknesses:**

The comparison on artificial networks is misleading. The authors should use benchmarks that create ground-truth networks generated in the hyperbolic space using the nonuniform popularity similarity model (nPSO) that allows also the creation of mesoscale community structure in artificial networks as many real networks have. Then they should use measures for the order of nodes on angular coordinates, the hyperbolic distance correlation and the angular separability index of the community. In particular, the presence of community in the benchmark is fundamental because they are a property that justifies why methods such as Hypermap and HypermapCN did not work well, because being based on the simple PSO model they did not have a parameter to model the community organization of real networks.

The test on real networks are not convincing because as a matter of fact the authors do not offer any quantitative evidence that these networks are hyperbolic and the measures they use are all empirical and ill posed. For instance, assuming that a network is embedded better because their greedy stretch factor is higher is misleading. In reality, every network has a complex system on the back with an intrinsic navigability and over-estimating it is wrong.

The measure of Greedy stretch factor for navigability is old and surpassed. Measures such as the greedy routing efficiency and the geometrical congruency are more advanced tools to quantify the navigability.

The authors do not consider in their comparison important studies:
+ one of the recent states of the art proposed in this article:
CLOVE, a Travelling Salesman’s approach to hyperbolic embeddings of complex networks with communities. SG Balogh, B Sulyok, T Vicsek, G Palla. Communications Physics 8 (1), 397
+ a fast method based on network automata:
Minimum curvilinear automata with similarity attachment for network embedding and link prediction in the hyperbolic space.A Muscoloni, CV Cannistraci. arXiv preprint arXiv:1802.01183
+ the studies of Filippo Radicchi on this topic seems to be neglected.
+ It does not seem that the Authors mentions explicitly the algorithms Hypermap and HypermapCN in their initial review. These are inefficient algorithm with low performance but they are between the first methods offered. Papadopoulos, F., Psomas, C. & Krioukov, D. Network mapping by replaying hyperbolic growth. IEEE/ACM Trans. Netw 23, 198–211 (2015).
Papadopoulos, F., Aldecoa, R. & Krioukov, D. Network geometry inference using common neighbors. Phys. Rev. E 92, 22807 (2015).
+  It does not seem that the Authors mentions explicitly the algorithms of LPCS of Wang, Z., Wu, Y., Li, Q., Jin, F. & Xiong, W. Link prediction based on hyperbolic mapping with community structure for complex networks. Phys. A 450, 609–623 (2016).
This is one of the fastest ever algorithm proposed, and it should be considered together with the other evolutions proposed by the same authors in subsequent articles that the Authors can find themselves by reviewing the literature of these relevant scientists.
7. This study seems to be neglected together with the following of the same authors: Martin Keller‑Ressel and Stephanie Nargang authored “HYDRA: a method for strain-minimizing hyperbolic embedding of network- and distance-based data”.

Coalescent embedding, which is a machine learning method, was published on arXiv on [Submitted on 21 Feb 2016, Machine learning meets network science: dimensionality reduction for fast and efficient embedding of networks in the hyperbolic space], hence before the other machine learning studies listed.

The study does not report enough information to replicate the experiments. For many algorithms, it is not reported the way their versions or hyperparameters are selected.

The organization of the article needs a strong effort to improve  the quality of the information reported.
For instance the authors could propose a first figure that organizes the different algorithms in a genealogical tree in order of date of publication and relationship of methodology used.
The main article could report the selected results for the best methods and the appendix the full results.

**Questions:**

I kindly ask to the authors to address all the concerns I raised in the section Weaknesses above.
However, in my opinion this study needs to be totally restructured and re-written and my recommendation  is to withdraw and resubmit in a next conference a majorly improved version of this study.

---

> ### Author Response · Authors · 2025-11-26
>
> > The measure of Greedy stretch factor for navigability is old and surpassed. Measures such as the greedy routing efficiency and the geometrical congruency are more advanced tools to quantify the navigability.
>
> Thank you for bringing GRE to our attention; we strongly agree that GSF has issues, and GRE fixes them. We have also extended our framework to report GRE. The previous omission was because greedy stretch factor (GSF) was more commonly reported in the NT research we compared.
>
> > The test on real networks are not convincing because as a matter of fact the authors do not offer any quantitative evidence that these networks are hyperbolic
>
> These networks have been taken from other papers we cite; in many cases, they allowed to introduce the main contributions of these papers. Furthermore, these papers provide explanations for their inclusion. Collecting different kinds of graphs allows us to compare embedders more broadly.
>
> > Then they should use measures for the order of nodes on angular coordinates, the hyperbolic distance correlation and the angular separability index of the community.
>
> We know the measures such as C-score, however, the comparison oriented at the angular coordinates is a subject for further research (since, as far as we know, it makes sense only for artificial networks generated from HRG, PSO or H2/S1 models). This study is aimed at distance-based measures, which seems a more prevalent case in the ML community.
>
> > one of the recent states of the art proposed in this article: CLOVE, a Travelling Salesman’s approach to hyperbolic embeddings of complex networks with communities. SG Balogh, B Sulyok, T Vicsek, G Palla. Communications Physics 8 (1), 397
>
> This paper has been published (in a peer-reviewed venue) after the ICLR2025 submission deadline. We have included CLOVE in our framework.
>
> > It does not seem that the Authors mention explicitly the algorithms Hypermap and HypermapCN in their initial review. These are inefficient algorithm with low performance but they are between the first methods offered. Papadopoulos, F., Psomas, C. & Krioukov, D. Network mapping by replaying hyperbolic growth. IEEE/ACM Trans. Netw 23, 198–211 (2015). Papadopoulos, F., Aldecoa, R. & Krioukov, D. Network geometry inference using common neighbors. Phys. Rev. E 92, 22807 (2015).
>
> We have mentioned these papers in the original revision (lines 181-182). Possibly the Reviewer has not found them because we do not refer to them by name, only by citation. We have added the names Hypermap and HypermapCN to the paper to make it more straightforward. As the Reviewer observes, these algorithms are inefficient, so we have decided not to include them in
> our framework (for now).
>
> > It does not seem that the Authors mention explicitly the algorithms of LPCS of Wang, Z., Wu, Y., Li, Q., Jin, F. & Xiong, W. Link prediction based on hyperbolic mapping with community structure for complex networks. Phys. A 450, 609–623 (2016). This is one of the fastest ever algorithm proposed, and it should be considered together with the other evolutions proposed by the same authors in subsequent articles that the Authors can find themselves by reviewing the literature of these relevant scientists.
>
> Indeed -- we mention an earlier $O(n^2)$ work by similar authors (line 182), but not this one. It seems we assumed this paper was just an application of hyperbolic embeddings rather than an actual proposal for an embedding method. Thank you for bringing this issue to our attention. We have added this algorithm to our framework.
>
> > the studies of Filippo Radicchi on this topic seems to be neglected.
>
> Which studies of Filippo Radicchi do you mean? His studies we have found seem to be more applications of embedders than new embedders themselves.
>
> > a fast method based on network automata: Minimum curvilinear automata with similarity attachment for network embedding and link prediction in the hyperbolic space.A Muscoloni, CV Cannistraci. arXiv preprint arXiv:1802.01183
>
> According to the abstract, "the embedding accuracy of this method seems superior to HyperMap-CN and inferior to coalescent embedding; however, its link prediction performance on real networks is without precedent for methods based on the hyperbolic space". Since we do not study the link prediction performance, we have decided not to include this embedder in our framework (we plan to do this in further research).

---

> ### Author Response · Authors · 2025-11-26
>
> > This study seems to be neglected together with the following of the same authors: Martin Keller‑Ressel and Stephanie Nargang authored “HYDRA: a method for strain-minimizing hyperbolic embedding of network- and distance-based data”.
>
> This method seems to be aimed at minimizing "strain," which is quite different in nature from the measures we study here. Additionally, similar to h-MDS, it significantly depends on a scaling parameter, which comes with no suggestions on how to obtain this (apart from grid-searching, which we rejected doing for any embedder not to cause unfair setups -- we limit the tinkering with hyperparameters and use the default/suggested setups by the Authors). In line with this decision, we ran it on small graphs (with default scaling), and its mAP and greedy routing results were bad, so we have decided not to include this embedder in our framework to compromise somehow (we might do this in further research).
>
> > The study does not report enough information to replicate the experiments. For many algorithms, it is not reported the way their versions or hyperparameters are selected.
>
> In line 296, we state that "we use the official implementations and hyperparameters". Unfortunately, there is not enough space in the main part of the article to say more. Our whole framework is included in the supplement, so the exact details could be seen there (which is better than some papers we cite, which include the code but not all the parameters necessary to replicate the experiments). However, we agree that it would be beneficial to include this information in the paper. Thus, we have updated Appendix A to explain our choices for hyperparameter values and other settings in detail.
>
> > The organization of the article needs a strong effort to improve the quality of the information reported. For instance, the authors could propose a first figure that organizes the different algorithms in a genealogical tree in order of date of publication and relationship of methodology used. The main article could report the selected results for the best methods and the Appendix the full results.
>
> Thank you for the visualization idea! While such a visualization would be interesting, we do not think it would be a proper addition to this paper. First, such a visualization would need a lot of space (and space constraints at ML conferences are scarce); second, given the number of papers in the field, it would also benefit from embedding in hyperbolic space, which is better suited to interactive setups. While we plan to prepare such an interactive visualization in the future,  with more feedback from the community, we believe it would better fit a survey paper or chapter than an experimental comparison paper.

---

### Author Response · Authors · 2025-11-26
**first revision**

We are grateful to all the Reviewers for their comments. They were all very useful for a substantially improved revision of the paper. We are still working on the final version; we attach the current revision to allow the referees to discuss whether our changes go in the right direction.

TL:DR

Important changes so far (in an arbitrary order):
- added CLOVE and LPCS embedders, and the GRE measure, to the framework
- an attempt to improve visualization (colored tables, aggregate plots, confidence intervals...)
- improved the literature review
- explanations regarding Lorentz and Poincare discrepancies
- some reorganization of the text
- more details on the setup in the appendix
- less reliance on dMAP / dMeanRank

Still planned:
- additional results regarding simulated networks (aggregate plots)
- possible further reorganization of texts, tables, and plots
- final proofreading

Details:

We thank Reviewer 1A68 for their literature suggestions. We have significantly revised the network theory literature review section; in particular, we have emphasized that there are many models used in the NT community, and included a mention (and, in the experimental section, evaluation) of the GR-score, which we feel is indeed better than the stretch factor. (Note: this measure is called Greedy Routing Efficiency in some sources (e.g., https://arxiv.org/pdf/1901.07909), so we assumed that the Reviewer meant that measure; however,
later we noticed that, in the CLOVE paper and in https://www.nature.com/articles/s41467-022-34634-6, GRE refers to the variant which compares geodesic distances to greedy piecewise linear paths, and we have not yet computed that measure.)

Based on Reviewer 1A68's suggestions, we have also included LPCS and CLOVE in the list of embedders we compare. In the case of CLOVE's omission, we note that it was published after the ICLR submission deadline, and our "unwritten" rule is that we do not include preprints, as they may still be changing before the actual acceptance. Our current results show it tends to achieve very good results in a very good time, both for hierarchies and networks, substantially changing the current state-of-the-art.

Based on another Reviewer 1A68's suggestion, we hope we have improved the paper's organization (we plan to improve it even further), and included more details on hyperparameters in Appendix A.

Reviewers 61kf and XXeR complain about the figures' low readability. Unfortunately, it is not easy to improve this due to the space limits and the large (and increasing after each round of reviews) number of embedders. That is why we have decided to include some new visualizations and reworked tables. We have also worked on the readability of tables in the Appendix, highlighting the best results for (hopefully) easier reading, and reporting the confidence intervals (instead of the results of all iterations) for repeated experiments.

Reviewer VCXG has asked about the reproducibility of Nickel and Kiela's results. The most serious non-reproducibility issue is the claimed poor efficiency of high-dimensional Euclidean embeddings. We have found a 2021 paper by Bansal and Benton which focuses on this specific non-reproducibility issue, so we added a citation to this paper.

Reviewer XXeR mentions the reliance on the discrete variants (dMAP / dMeanRank). We used the discrete variants mainly to evaluate the DHRG embedder, which relies on them; other embedders were evaluated using continuous measures. We agree that this was a somewhat ill-motivated decision, so we have instead switched to evaluating DHRG results using the continuous variants of the measures, similar to how it was done in the DHRG paper. We keep a mention of dMAP / dMeanRank in the main paper (its advantage is not only numerical precision but also fast computation), but shift reporting the obtained values of these measures to the Appendix.

---

### Author Response · Authors · 2025-12-04
**second revision**

We are very grateful for the Reviewers, who have made many great suggestions on how to improve the paper (and we really mean it,
this was by far the best experience we had in our recent submissions). It took us some time to implement these suggestions, and we
have published our responses relatively late -- and the reviewers had no chance to respond before the surprise freezing.

We have created a second revision which implements the "still planned" changes from the first revision.
We have also decided to add two additional classic benchmark networks (openflights and ias), reimplemented LPCS in C++
and fixed some bugs (the most serious bug caused misreporting some quality measures for some embedders on the csphd dataset
due to a large number of vertices of degree 0). We have also added a citation to F. Radicchi's work that Reviewer 1A68 probably meant.

---

### Meta-Review · Area_Chair_h1W3 · 2025-12-14

**Summary:**

This paper empirically compare hyperbolic and related embedding across communities. This is the strength of the paper -- comprehensive and fair comparison with practical take-away and a reflection of the literature. There are new hyperbolic embedding emerging each year, and this work is potentially a more valuable effort to assess these embeddings with multiple performance metrics (including the authors' own ICV). The raised weaknesses is *not methodological*, but on whether static embedding is meaningful and lack of theory. The criticism by Reviewer 1A68 (most critic reviewer) is on lack of baseline methods and performance metrics from the network theory community. The author's rebuttal addresses these weaknesses by enriching the literature, and adding baselines and metrics (e.g., GRE). The paper is well-written --- as the authors mentioned, the paper has gone through iterations. Overall, I recommend acceptance based on the empirical comprehensiveness and potential significance to spark inter-community discussions. In preparing the final version, please make sure the limitations are discussed explicitly as mentioned by R-1A68, R-VCXG and other reviewers.

**Reviewer Concerns:**

R-1A68: missing baselines/metrics and literature. The authors have added more baselines/metrics and enriched the literature.

R-VCXG: static-embedding is less relevant without downstream tasks. The authors clarified that the Poincare embedding is also static and one can swap embeders in ML pipelines.

The other reviewers are mostly positive, or whose comments have been addressed in the rebuttal so that they would flip to the acceptance side.

**Reviewer Scores:**

The most critic reviewers R-1A68 and R-VCXG should increase their scores, although R-1A68 may still be on the rejection side, as the authors have comprehensively addressed their comments. After the discussion, the other reviewers would be on the acceptance side. Overall, the authors made a good rebuttal leading to an acceptance case.

---

> ### Public Comment · ~Eryk_Kopczyński1 · 2026-02-28
>
> Thanks for the acceptance!
>
> In the camera ready version:
> * We have included an explicit mention of the limitations (as suggested by the meta review) in the Conclusions and Abstract.
> * We have included the HMCS embedder in the framework. We believe this embedder should definitely be included; Reviewer R-1A68 did not name this embedder directly but they probably meant this embedder in  "should be considered together with the other evolutions proposed by the same authors in subsequent articles". This did not affect the results significantly.
> * We have included temperature T=0.9 in our experiments on simulated networks, which did not affect the conclusion. We have also added a mention of other possible extensions of the setup (different power-law coefficients, different generative models used) in the Conclusion.
> * Our public GitHub repository is now updated to include the additions written for this paper; we have provided a link and also updated the supplementary material to use the new version.
> * We have fixed some minor issues that have been introduced during our (very intensive) work during the discussion phase: missing figures, language correction, visibility, etc.

---

> > ### Public Comment · ~Eryk_Kopczyński1 · 2026-05-09
> > **post-conference update**
> >
> > We have decided to use the opportunity to re-upload after the conference to fix some minor issues we found during preparing the poster and discussions at the conference.
> >
> > For completeness, here are the issues fixed:
> > - `Human2`, `Human8` and `Macaque4` connectomes were loaded incorrectly
> > - added some missing results (HMCS on the `yeast` graph, TreeRep on the `astroph` and `hepph` graphs)
> > - fixed the visualize script (`scripts/visualize.sh`)
> > - we declared also posting full data to FigShare, but we did not actually do so -- this is fixed

---

### Decision · Program_Chairs · 2026-01-26

Accept (Poster)